# NoiseDiffusion:
# Correcting Noise for Image Interpolation with Diffusion Models beyond Spherical Linear Interpolation

**Pengfei Zheng**[1]   **Yonggang Zhang**[2]   **Zhen Fang**[3]   **Tongliang Liu**[4]   **Defu Lian**[1]*   **Bo Han**[2]
[1]University of Science and Technology of China   [2]TMLR Group, Hong Kong Baptist University
[3]University of Technology Sydney   [4]Sydney AI Centre, The University of Sydney

## Abstract

Image interpolation based on diffusion models is promising in creating fresh and interesting images. Advanced interpolation methods mainly focus on spherical linear interpolation, where images are encoded into the noise space and then interpolated for denoising to images. However, existing methods face challenges in effectively interpolating natural images (not generated by diffusion models), thereby restricting their practical applicability. Our experimental investigations reveal that these challenges stem from the invalidity of the encoding noise, which may no longer obey the expected noise distribution, e.g., a normal distribution. To address these challenges, we propose a novel approach to correct noise for image interpolation, *NoiseDiffusion*. Specifically, NoiseDiffusion approaches the invalid noise to the expected distribution by introducing subtle Gaussian noise and introduces a constraint to suppress noise with extreme values. In this context, promoting noise validity contributes to mitigating image artifacts, but the constraint and introduced exogenous noise typically lead to a reduction in signal-to-noise ratio, i.e., loss of original image information. Hence, NoiseDiffusion performs interpolation within the noisy image space and injects raw images into these noisy counterparts to address the challenge of information loss. Consequently, NoiseDiffusion enables us to interpolate natural images without causing artifacts or information loss, thus achieving the best interpolation results. Our code is available at https://github.com/tmlr-group/NoiseDiffusion.

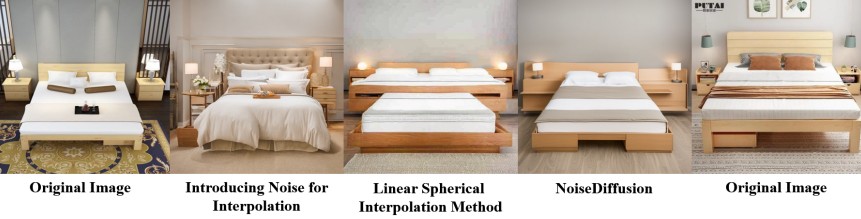

| Original Image | Introducing Noise for Interpolation | Linear Spherical Interpolation Method | NoiseDiffusion | Original Image |

Figure 1: Comparison of images generated with different interpolation methods.

## 1 Introduction

Image interpolation is an exceptionally fascinating task, not only for generating analogous images but also for igniting creative applications, especially in domains like advertising and video generation. At present, state-of-the-art generative models showcase the ability to produce intricate and captivating visuals, with many recent breakthroughs deriving from diffusion models (Ho et al., 2020; Song et al., 2021a; Rombach et al., 2022; Saharia et al., 2022b; Ramesh et al., 2022). The potential

---

*Corresponding Author Defu Lian (liandefu@ustc.edu.cn)

of diffusion models is widely acknowledged, but to our knowledge, there has been relatively little research on image interpolation with diffusion models (Croitoru et al., 2023).

Within the realm of diffusion models, the prevailing technique for image interpolation is spherical linear interpolation (Song et al., 2021a;b). This approach shines when employed with images generated by diffusion models. However, when extrapolated to natural images, the quality of interpolation results might fall short of expectations and frequently introduce artifacts, as depicted in Figure 2.

We initially analyze the spherical linear interpolation process and attribute subpar interpolation results to the invalidity of the encoding noise. This noise does not obey the expected normal distribution and may contain noise components at levels higher or lower than the denoising threshold[1], resulting in artifacts in the final interpolated images. Directly manipulating the mean and variance of the noise through translation and scaling is a straightforward approach to bring it closer to the desired distribution. However, this not only fails to improve the image quality but also results in the loss of image information. In addition, combined with the SDEdit method(Meng et al., 2022), we directly introduce standard Gaussian noise for interpolation. While this method improves the quality of images, it comes at the expense of introducing additional information, as depicted in Figure 4.

To improve the interpolation results, we propose a novel approach to correct noise for image interpolation, *NoiseDiffusion*. Specifically, NoiseDiffusion approaches the invalid noise to the expected distribution by introducing subtle Gaussian noise and introduces a constraint to suppress noise with extreme values. In this context, promoting noise validity contributes to mitigating image artifacts, but the constraint and introduced exogenous noise typically lead to a reduction in signal-to-noise ratio, i.e., loss of original image information. Hence, NoiseDiffusion subsequently performs interpolation in the noisy image space and injects raw images into these noisy images to tackle the information loss issue. These enhancements enable us to interpolate with natural images without artifacts, yielding the best interpolation results achieved to date. Considering the limited exploration of previous research in this field (Croitoru et al., 2023), we hope that our research can provide inspiration for future research.

## 2 RELATED WORK

**Diffusion Models** Diffusion models create samples from the Gaussian noise using sequential denoising steps. To date, diffusion models have been applied to various tasks, including image generation (Rombach et al., 2022; Song & Ermon, 2020; Nichol et al., 2022; Jiang et al., 2022), image super-resolution (Saharia et al., 2022c; Batzolis et al., 2021; Daniels et al., 2021), image inpainting (Esser et al., 2021), image editing (Meng et al., 2022), and image-to-image translation (Saharia et al., 2022a). In particular, latent diffusion models (Rombach et al., 2022) excel in generating text-conditioned images, receiving widespread acclaim for their ability to produce realistic images.

**Image Interpolation** Earlier approaches, such as StyleGAN (Karras et al., 2019), allowed for interpolation using the latent variables of images. However, their effectiveness is constrained by the model's ability to represent only a subset of the image manifold, presenting challenges when applied to natural images (Xia et al., 2022). What's more, latent diffusion models can utilize prompts to interpolate the generated images (like Lunarring), but its interpolation potential on natural images has not yet been discovered. To the best of our knowledge, a method for interpolating natural images using latent variables with diffusion models has not been encountered.

## 3 PRELIMINARIES

In this section, we first introduce how to describe the diffusion model's noise injection and denoising process in the form of stochastic differential equations (SDEs). Building upon this, we provide a brief overview of how diffusion models are used for image interpolation and editing. Through image editing, we can implement an interpolation method that doesn't require latent variables, that is, introducing Gaussian noise and then denoising. These methods form the foundation of the proposed approach, NoiseDiffusion.

---

[1]Noise level of denoising

## 3.1 THE DETAILS OF DIFFUSION MODELS

**Perturbing Data With SDEs** (Song et al., 2021b) We denote the distribution of training data as $p_{\text{data}}(\boldsymbol{x})$, and the Gaussian perturbations applied to $p_{\text{data}}(\boldsymbol{x})$ by the diffusion model can be described by the following stochastic differential equation expression:

$$d\boldsymbol{x}_t = \boldsymbol{\mu}(\boldsymbol{x}_t, t)dt + \sigma(t)d\boldsymbol{w}_t, \tag{1}$$

where $t \in [0, T], T > 0$ is a fixed constant, $\{\boldsymbol{w}_t\}_{t \in [0,T]}$ denotes the standard Wiener process (a.k.a., Brownian motion), $\boldsymbol{\mu}(\cdot, \cdot) : \mathbb{R}^d \to \mathbb{R}^d$ is a vector-valued function called the drift coefficient of $\boldsymbol{x}_t$, and $\sigma(\cdot) : \mathbb{R} \to \mathbb{R}$ is a scalar function known as the diffusion coefficient.

We denote the distribution of $\boldsymbol{x}_t$ as $p_t(\boldsymbol{x}_t)$ and consequently, $p_0$ represents for the training data distribution $p_{\text{data}}$ and $p_T$ is an unstructured prior distribution that contains no information of $p_0$.

**Generating Samples By Reversing the SDEs** (Song et al., 2021b) By starting from samples of $p_T$ and reversing the perturbation process, we can obtain samples $\boldsymbol{x}_0 \sim p_0$. The reverse of a diffusion process is also a diffusion process and can be given by the reverse-time SDE (Anderson, 1982):

$$d\boldsymbol{x} = [\boldsymbol{\mu}(\boldsymbol{x}_t, t) - \sigma(t)^2 \nabla \log p_t(\boldsymbol{x}_t)]dt + \sigma(t)d\bar{\boldsymbol{w}}, \tag{2}$$

where $\bar{\boldsymbol{w}}$ is a standard Wiener process when time flows backwards from $T$ to $0$, and $dt$ is an infinitesimal negative timestep. Once the score of each marginal distribution, $\nabla \log p_t(\boldsymbol{x})$, is known for all $t$, we can derive the reverse diffusion process from Eq.2 and simulate it to sample from $p_0$. And methods like stochastic Runge-Kutta (Kloeden et al., 1992) methods can be used to solve this.

**Probability Flow ODE** (Song et al., 2021b) Diffusion models enable another numerical method for solving the reverse-time SDE. For all diffusion processes, there exists a corresponding deterministic process whose trajectories share the same marginal probability densities $\{p_t(\boldsymbol{x}_t)\}_{t=0}^T$ as the SDE. This deterministic process satisfies an ordinary differential equation (ODE) :

$$d\boldsymbol{x}_t = [\boldsymbol{\mu}(\boldsymbol{x}_t, t) - \frac{1}{2}\sigma(t)^2 \nabla \log p_t(\boldsymbol{x}_t)]dt, \tag{3}$$

which can be determined from the SDE once scores are known. Usually we call the ODE in Eq.3 the probability flow ODE.

## 3.2 IMAGE EDITING

**Spherical Linear Interpolation** In diffusion models, the prevailing image interpolation method is spherical linear interpolation (Song et al., 2021a;b):

$$\boldsymbol{x}_T^{(\lambda)} = \frac{\sin((1-\lambda)\theta)}{\sin(\theta)}\boldsymbol{x}_T^{(0)} + \frac{\sin(\lambda\theta)}{\sin(\theta)}\boldsymbol{x}_T^{(1)},$$

where $\theta = \arccos(\frac{(\boldsymbol{x}_T^{(0)})^\mathsf{T}\boldsymbol{x}_T^{(1)}}{\|\boldsymbol{x}_T^{(0)}\|\|\boldsymbol{x}_T^{(1)}\|})$, and $\lambda$ is a coefficient that controls interpolation style between two images. $\boldsymbol{x}_T^{(i)}$ can be either a noisy image encoded from image $\boldsymbol{x}_0^{(i)}$ by integrating Eq.3, or randomly sampled standard Gaussian noise. After completing the interpolation of latent variables through the above equation, decoding can be achieved by integrating the corresponding ODE for the reverse-time SDE. In the rest of the paper, we use $\texttt{slerp}(\boldsymbol{x}_t^{(0)}, \boldsymbol{x}_t^{(1)}, \lambda)$ to denote the spherical linear interpolation of the latent variables $\boldsymbol{x}_t^{(0)}$ and $\boldsymbol{x}_t^{(1)}$ with the interpolation coefficient $\lambda$.

**Image Editing with SDEdit (Meng et al., 2022)** The SDEdit accomplishes image modifications by overlaying the desired alterations onto the image, introducing noise, and subsequently denoising the composite. This process ensures that the resulting image maintains a high level of quality. For any given image $\boldsymbol{x}_0$, the SDEdit procedure is defined as follows:

Sample $\boldsymbol{x}_t \sim \mathcal{N}(\boldsymbol{x}_0; \sigma^2(t_0)\boldsymbol{I})$, then produce $\hat{\boldsymbol{x}}_0$ by solving Eq.2.

For appropriately trained SDE models, a trade-off between realism and faithfulness emerges when varying the values of $t_0$. When we add more Gaussian noise and run the SDE for longer, the synthesized images are more realistic but less faithful. Conversely, adding less Gaussian noise and running the SDE produces synthesized images that are more faithful but less realistic.

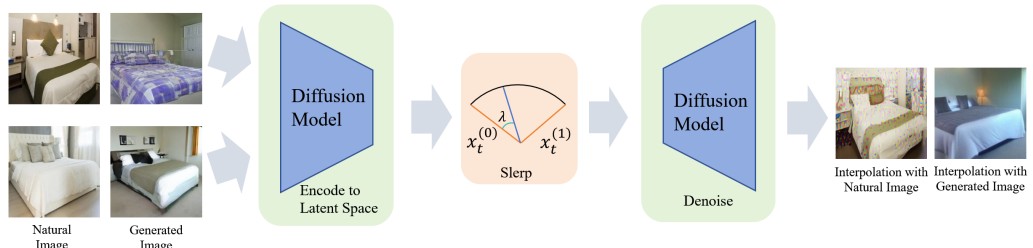

Figure 2: The spherical linear interpolation. Original images: The images on the left are natural images, whereas the images on the right are generated by the diffusion model. Interpolation results: The images on the left and right are the interpolation results of natural images and images generated by diffusion model respectively.

# 4 THE IMAGE INTERPOLATION METHODS

## 4.1 THE SPHERICAL LINEAR INTERPOLATION OF IMAGES

Let's start by introducing the process of spherical linear interpolation of images. Given two images, the initial step involves encoding them into a latent space, i.e., Eq. 4 and 5. Then, we can perform spherical linear interpolation on the latent variables, i.e., Eq. 6, followed by denoising to generate the interpolation results with Eq. 7.

$$\boldsymbol{x}_t^{(0)} = \boldsymbol{f}(\boldsymbol{x}_0^{(0)}, t), \tag{4}$$

$$\boldsymbol{x}_t^{(1)} = \boldsymbol{f}(\boldsymbol{x}_0^{(1)}, t), \tag{5}$$

$$\boldsymbol{x}_t = \texttt{slerp}(\boldsymbol{x}_t^{(0)}, \boldsymbol{x}_t^{(1)}, \lambda), \tag{6}$$

$$\hat{\boldsymbol{x}}_0 = \boldsymbol{f}^{-1}(\boldsymbol{x}_t, t). \tag{7}$$

In this context, we denote the Gaussian noise as $\epsilon_t \sim \mathcal{N}(\mathbf{0}, \sigma(t)^2 \boldsymbol{I})$ and the original image as $\boldsymbol{x}_0^{(i)}$ with $i \in \{0, 1\}$, respectively. Accordingly, $\boldsymbol{x}_t^{(i)}$ represents the noisy image corresponding to the variable of the image in the latent space with noise level $\sigma(t)$. Using the probability flow ODE for its stability and unique encoding capabilities, we encode $\boldsymbol{x}_0$ into the latent space by integrating Eq.3, and we denote this encoding process as a function $\boldsymbol{f}$. Similarly, we denote the decoding process as $\boldsymbol{f}^{-1}$, which corresponds to denoising through the ODE associated with the reverse-time SDE.

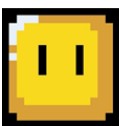 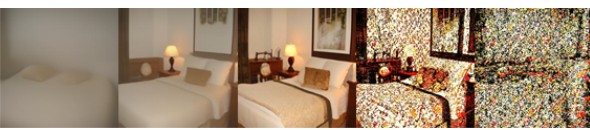

Figure 3: The impact of noise levels. We added Gaussian noise with levels of $\sigma(t) = [70, 75, 80, 85, 90]$ to the image on the left. Subsequently, we applied denoising to each noisy image with the same noise level of $\sigma(t') = 80$, resulting in the denoised images on the right.

Examining Figure 2, we notice that the interpolation result derived from natural images (not generated from diffusion model) displays noticeable artifacts, contrasting with the one derived from images generated by the diffusion model, which is free from such imperfections.

## 4.2 THE REASON FOR FAILURE

To explore what kind of potential variables can be better denoised, we add Gaussian noise to the image at various noise levels $\sigma(t)$, resulting in $\boldsymbol{x}_t = \boldsymbol{x}_0 + \epsilon_t$, and then denoise them at the same noise level $\sigma(t')$, yielding $\hat{\boldsymbol{x}}_0 = \boldsymbol{f}^{-1}(\boldsymbol{x}_t, t')$. The results are shown in Figure 3.

Based on the results depicted in Figure 3, we observe that adding Gaussian noise matching the denoising level produces high-quality images. However, when the noise level exceeds the denoising

threshold, additional artifacts are introduced in the generated images. Conversely, when the noise level falls below the denoising threshold, the resulting images appear somewhat blurred, accompanied by a noticeable loss of features.

This phenomenon is rather peculiar since, in the context of a Gaussian distribution, points closer to the mean typically exhibit higher probability density. In other words, within the framework of the diffusion model, noisy images with lower noise levels (closer to the mean) should ideally be more effectively denoised. Building upon these observations, we introduce Theorem 1 to provide an explanation for this phenomenon:

**Theorem 1.** *The standard normal distribution $\mathcal{N}(\mathbf{0}, \boldsymbol{I}_n)$ in high dimensions is close to the uniform distribution on the sphere of radius $\sqrt{n}$.*

The detailed proof process of Theorem 1 can be found in Appendix A.1. Theorem 1 indicates that random variables following the standard normal distribution in high dimensions are primarily distributed on a hypersphere. This is because, as we approach the mean, the probability density increases, but the volume in high-dimensional space gradually expands as we move away from the mean. This result neatly explains why only noisy images with noise levels matching the denoising threshold can produce high-quality results after denoising: During the training process, the model can only observe noisy images primarily reside on the hypersphere. Consequently, it can only effectively recover images of this nature.

Building upon Theorem 1, we can attribute the failure of spherical linear image interpolation to the mismatch between noise levels and denoising threshold. The natural images encompass numerous features that the model has not previously encountered. Consequently, the latent variables do not obey the expected normal distribution, and may contain noise components at levels higher or lower than the denoising threshold, resulting in low image quality after denoising. Inspired by SDEdit, we can directly introduce Gaussian noise to the images as a solution to this mismatch problem. Details are as follows.

### 4.3 INTRODUCING NOISE FOR INTERPOLATION

Here, we introduce the image interpolation method combined with SDEdit. When given two images, the method starts by introducing Gaussian noise at the same level to each of them. Following this, we employ spherical linear interpolation and subsequently apply denoising:

$$\boldsymbol{x}_t^{(0)} = \boldsymbol{x}_0^{(0)} + \epsilon_t, \tag{8}$$

$$\boldsymbol{x}_t^{(1)} = \boldsymbol{x}_0^{(1)} + \epsilon_t, \tag{9}$$

$$\boldsymbol{x}_t = \texttt{slerp}(\boldsymbol{x}_t^{(0)}, \boldsymbol{x}_t^{(1)}, \lambda), \tag{10}$$

$$\hat{\boldsymbol{x}}_0 = \boldsymbol{f}^{-1}(\boldsymbol{x}_t, t). \tag{11}$$

The noise added to the images can be either the same or different. Shortly, we will demonstrate that they exhibit only minor distinctions. However, it is crucial to emphasize that since this image interpolation method is based on SDEdit, it unavoidably inherits the drawbacks of the SDEdit method, as illustrated in Figure 4.

The interpolation results presented in Figure 4 indicate that the method can address the issue of poor image quality. However, when we add more Gaussian noise and denoise, the interpolated images, while maintaining the original style, exhibit a phenomenon resembling direct image overlay. Conversely, selecting less Gaussian noise and denoising, while ensuring realistic images, introduces additional information, ultimately resulting in interpolation failure.

### 4.4 NOISEDIFFUSION

Based on the experimental results above, we can conclude the following: when spherical linear interpolation is directly applied to natural images, the resulting images can better preserve the original features but may contain artifacts. Conversely, directly introducing noise for image interpolation may yield high-quality images but often causes the information loss issue. To integrate these two methods, we propose the following theorem.

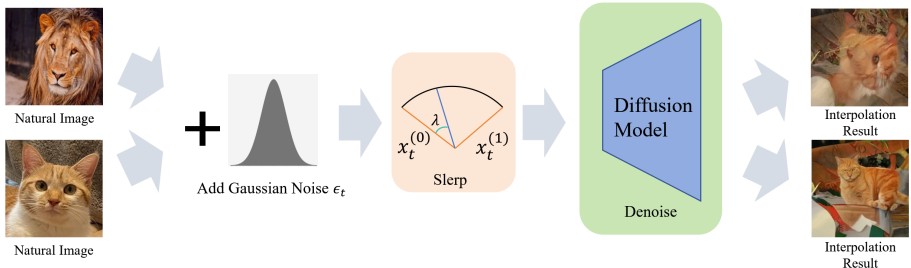

Figure 4: Introducing noise for image interpolation. In the interpolated images, the top one represents the interpolation result with less Gaussian noise, while the bottom one represents the interpolation result with more Gaussian noise.

**Theorem 2.** *In high-dimensional spaces, independent and isotropic random vectors tend to be almost orthogonal.*

The detailed proof process of Theorem 2 can be found in Appendix A.2. Based on Theorem 1 and Theorem 2, we proposed a new image interpolation method called NoiseDiffusion: Given two images, we begin by encoding them into the latent space and clip them to suppress noise with extreme values. Next, we synthesize the latent variables with Gaussian noise, combining them with the original images, and finally apply clipping and denoising to produce the interpolation results:

$$\boldsymbol{x}_t^{(0)} = \texttt{clip}(\boldsymbol{f}(\boldsymbol{x}_0^{(0)}, t)), \tag{12}$$

$$\boldsymbol{x}_t^{(1)} = \texttt{clip}(\boldsymbol{f}(\boldsymbol{x}_0^{(1)}, t)), \tag{13}$$

$$\boldsymbol{x}_t = \alpha * \boldsymbol{x}_t^{(0)} + \beta * \boldsymbol{x}_t^{(1)} + (\mu - \alpha) * \boldsymbol{x}_0^{(0)} + (\nu - \beta) * \boldsymbol{x}_0^{(1)} + \gamma * \epsilon_t, \tag{14}$$

$$\hat{\boldsymbol{x}}_0 = \boldsymbol{f}^{-1}(\texttt{clip}(\boldsymbol{x}_t), t). \tag{15}$$

In these equations, $\alpha$ and $\beta$ correspond to coefficients for image style, while $\mu$ and $\nu$ serve as compensation coefficients to adjust the amount of original image information. Additionally, $\gamma$ represents the lubrication coefficient, which can be used to adjust the amount of noise to enhance image quality.

Ensuring that the formula $\sqrt{\alpha^2 + \beta^2 + \gamma^2} = 1$ is satisfied is crucial. Drawing from Theorem 1 and Theorem 2, we can infer that for any three high-dimensional vectors on a hypersphere with a radius of $\|r\|$, denoted as $\boldsymbol{v}_1$, $\boldsymbol{v}_2$ and $\boldsymbol{v}_3$, the magnitude of the weighted sum $\boldsymbol{v}_{12}$ is given by $\|\boldsymbol{v}_{12}\| = \|\alpha \cdot \boldsymbol{v}_1 + \beta \cdot \boldsymbol{v}_2\| = \sqrt{\alpha^2 \|\boldsymbol{v}_1\|^2 + \beta^2 \|\boldsymbol{v}_2\|^2 + 2\|\boldsymbol{v}_1\|\|\boldsymbol{v}_2\| \cos\theta} = \sqrt{\alpha^2 + \beta^2}\|r\|$. Moreover, it is worth noting that the newly obtained vector $\boldsymbol{v}_{12}$ and the vector $\boldsymbol{v}_3$ also remain orthogonal. Consequently, we can represent the magnitude of the weighted sum of these vectors as: $\|\alpha \cdot \boldsymbol{v}_1 + \beta \cdot \boldsymbol{v}_2 + \gamma \cdot \boldsymbol{v}_3\| = \sqrt{\alpha^2 + \beta^2 + \gamma^2}\|r\|$. While the denoised image in Figure 2 displays some artifacts, the majority of its content remains clear. This observation implies that the latent variables of natural images $\boldsymbol{v}_1 = \boldsymbol{x}_t^{(0)}$, $\boldsymbol{v}_2 = \boldsymbol{x}_t^{(1)}$ also tend to be near the hypersphere. Therefore, considering that Gaussian noise $\boldsymbol{v}_3 = \epsilon_t$ also resides on the hypersphere, it is crucial to maintain the formula $\sqrt{\alpha^2 + \beta^2 + \gamma^2} = 1$ to ensure that the final synthesized latent variable also possesses the same properties.

## 4.5 BOUNDARY CONTROL

According to the widely recognized statistical principle known as the **empirical rule** (also known as **68–95–99.7 rule**) (Pukelsheim, 1994), which pertains to the behavior of data within a normal distribution, approximately 99.7% of data points are located within three standard deviations from the mean. Consequently, considering our analysis of how noise above the denoising threshold impacts images, data points exhibiting significant deviations from the mean are considered potential sources of image artifacts, a hypothesis that will be validated in subsequent experiments. To mitigate their influence, we employ the following boundary control (clip) procedure :

$$\text{Pixel Value} = \begin{cases} \text{Boundary}, & \text{if Pixel Value} > \text{Boundary}, \\ -\text{Boundary}, & \text{if Pixel Value} < -\text{Boundary}, \\ \text{Pixel Value}, & \text{otherwise}. \end{cases}$$

### 4.6 THE CONNECTION OF METHODS

Here, we establish the relationship between our approach and other methods, highlighting the advantages of our approach. To begin with, our method, when coupled with appropriate parameter choices, can be adapted into two other methods:

**Spherical Linear Interpolation** Combining Theorem 2, as high-dimensional random vectors are orthogonal, we can express spherical linear interpolation in the following form with $\theta = \frac{\pi}{2}$:

$$\boldsymbol{x}_t^{(\lambda)} = \frac{\sin((1-\lambda)\theta)}{\sin(\theta)}\boldsymbol{x}_t^{(0)} + \frac{\sin(\lambda\theta)}{\sin(\theta)}\boldsymbol{x}_t^{(1)} = \sin((1-\lambda)\cdot\frac{\pi}{2})\boldsymbol{x}_t^{(0)} + \sin(\lambda\cdot\frac{\pi}{2})\boldsymbol{x}_t^{(1)}.$$

This is equivalent to our method with $\gamma = 0$, $\mu = \alpha = \sin((1-\lambda)\cdot\frac{\pi}{2})$, and $\nu = \beta = \sin(\lambda\cdot\frac{\pi}{2})$.

**Introducing Noise for Interpolation** We classify the method of introducing noise for image interpolation into two categories, assuming that the noise level is substantially higher than that of the image, which is often the common case. In this scenario, we can show that our approach can be adapted into this method:

1. The noise added to the image is the same:

$$\boldsymbol{x}_t = \mathtt{slerp}(\boldsymbol{x}_t^{(0)}, \boldsymbol{x}_t^{(1)}, \lambda) = \frac{\sin((1-\lambda)\cdot 0)}{\sin(0)}\boldsymbol{x}_t^{(0)} + \frac{\sin(\lambda\cdot 0)}{\sin(0)}\boldsymbol{x}_t^{(1)}$$

$$= (1-\lambda)\boldsymbol{x}_t^{(0)} + \lambda\boldsymbol{x}_t^{(1)} = (1-\lambda)\boldsymbol{x}_0^{(0)} + \lambda\boldsymbol{x}_0^{(1)} + (1-\lambda+\lambda)\epsilon_t$$

$$= (1-\lambda)\boldsymbol{x}_0^{(0)} + \lambda\boldsymbol{x}_0^{(1)} + \epsilon_t.$$

   This is equivalent to our method with $\alpha = \beta = 0, \mu = 1-\lambda, \nu = \lambda$.

2. The noise added to the image is different:

$$\boldsymbol{x}_t = \mathtt{slerp}(\boldsymbol{x}_t^{(0)}, \boldsymbol{x}_t^{(1)}, \lambda) = \frac{\sin((1-\lambda)\cdot\frac{\pi}{2})}{\sin(\frac{\pi}{2})}\boldsymbol{x}_t^{(0)} + \frac{\sin(\lambda\cdot\frac{\pi}{2})}{\sin(\frac{\pi}{2})}\boldsymbol{x}_t^{(1)}$$

$$= \sin((1-\lambda)\cdot\frac{\pi}{2})\boldsymbol{x}_0^{(0)} + \sin(\lambda\cdot\frac{\pi}{2})\boldsymbol{x}_0^{(1)} + \sin((1-\lambda)\cdot\frac{\pi}{2})\epsilon_t^{(0)} + \sin(\lambda\cdot\frac{\pi}{2})\epsilon_t^{(1)}$$

$$= \sin((1-\lambda)\cdot\frac{\pi}{2})\boldsymbol{x}_0^{(0)} + \sin(\lambda\cdot\frac{\pi}{2})\boldsymbol{x}_0^{(1)} + \epsilon_t'.$$

   This is equivalent to our method with $\alpha = \beta = 0, \mu = \sin((1-\lambda)\cdot\frac{\pi}{2}), \nu = \sin(\lambda\cdot\frac{\pi}{2})$.

Compared with spherical linear interpolation, our method introduces Gaussian noise to better position latent variables on the hypersphere. In contrast to the approach of introducing noise for interpolation, our method incorporates noise correction, which enables us to position latent variables on the hypersphere and remove artifacts with a smaller amount of Gaussian noise.

## 5 EXPERIMENTS

The SDE is typically designed such that $p_T$ is close to a tractable Gaussian distribution $\pi(\boldsymbol{x})$. We hereafter adopt the configurations in Karras et al. (2022), who set $\boldsymbol{\mu}(\boldsymbol{x}, t) = 0$ and $\sigma(t) = \sqrt{2t}$. In this case, we have $p_t(\boldsymbol{x}) = p_{\mathrm{data}}(\boldsymbol{x}) \otimes \mathcal{N}(\boldsymbol{0}, t^2\boldsymbol{I})$, where $\otimes$ denotes the convolution operation, and $\pi(\boldsymbol{x}) = N(\boldsymbol{0}, T^2\boldsymbol{I})$. We conduct evaluations on diffusion models trained on LSUN Cat-256 and LSUN Bedroom-256 images as a basis for our evaluation. We verify the effectiveness of our method on the Stable Diffusion (Rombach et al., 2022), and the results are provided in the Appendix C.

### 5.1 THE LUBRICATING COEFFICIENT

We keep all other parameters unchanged and incrementally increase $\gamma$ from 0 to 1, as illustrated in Figure 5. Upon observation, it is apparent that as $\gamma$ increases, the artifacts in the image gradually diminish, resulting in a notable enhancement in image quality. However, at the same time, the image gradually loses some of its original features and introduces additional information.

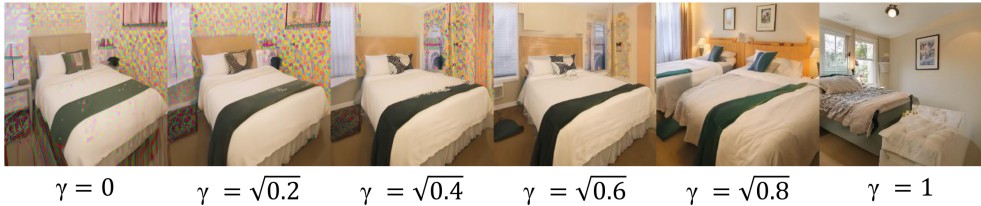

$$\gamma = 0 \qquad \gamma = \sqrt{0.2} \qquad \gamma = \sqrt{0.4} \qquad \gamma = \sqrt{0.6} \qquad \gamma = \sqrt{0.8} \qquad \gamma = 1$$

Figure 5: The impact of lubricating coefficient $\gamma$.

## 5.2 THE CHANGE IN STYLE

As shown in Figure 6, we can change the style of images by modifying the values of $\alpha$ and $\beta$. In order to facilitate comparison with the results of spherical linear interpolation, we choose $\alpha = \sin\left(\frac{\pi}{2} \cdot \lambda\right)$, $\beta = \cos\left(\frac{\pi}{2} \cdot \lambda\right)$ and $\gamma = 0$. Additionally, more interpolation results are available in the appendix (Figure 11 and Figure 12).

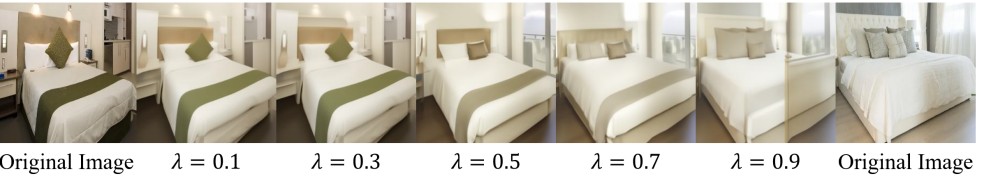

Original Image $\qquad$ $\lambda = 0.1$ $\qquad$ $\lambda = 0.3$ $\qquad$ $\lambda = 0.5$ $\qquad$ $\lambda = 0.7$ $\qquad$ $\lambda = 0.9$ $\qquad$ Original Image

Figure 6: The image style changes with the variation of $\lambda$.

## 5.3 BOUNDARY CONTROL

We implemented boundary control on the latent variables, and the results are depicted in Figure 8. It can be seen that as the boundaries decrease, the artifacts on the image are greatly reduced, which substantially improves the quality of the images. Furthermore, we compared three boundary control methods: control before interpolation, control after interpolation, and control before and after interpolation. The results of the three methods are shown in Figure 10. Upon examination, it can be observed that the method of applying constraints to latent variables before and after interpolation is more effective in reducing artifacts.

However, reducing the boundaries also leads to some loss of image features and darkening, which implies that boundary control can result in information loss. To address this issue, one effective strategy is to incorporate the original image information in the noisy image space, as detailed below.

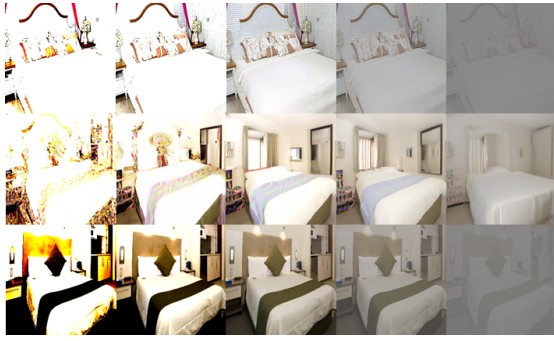

Figure 7: The effect of image scaling is demonstrated in the top and bottom rows of images, showcasing results obtained by scaling the original image with scales $l = [10, 2, 1, 0.5, 0.1]$. In the middle row, we present the outcomes of image interpolation, maintaining all parameters except for $\mu$ and $\nu$, which are adjusted to $\mu = \nu = [10, 2, 1, 0.5, 0.1]$.

## 5.4 THE IMPACT OF IMAGE INFORMATION

Figure 7 illustrates the impact of modifying the information of the original images (i.e., modifying the values of $\mu$ and $\nu$) on the interpolation results. It can be observed that smaller values of $\mu$ and $\nu$ lead to darker images while increasing them results in overly bright pictures. Images obtained with

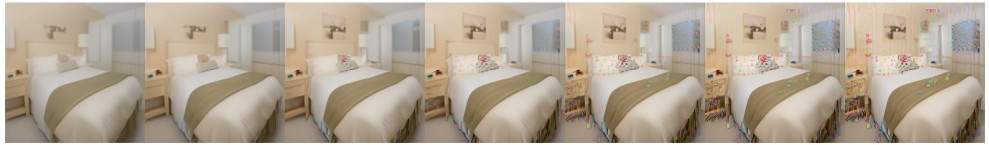

Boundary=2.0  Boundary=2.2  Boundary=2.4  Boundary=2.6  Boundary=2.8  Boundary=3.0  Boundary=3.2

Figure 8: The impact of the boundary.

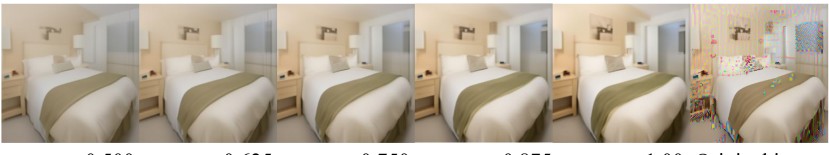

$\mu = \nu = 0.500$  $\mu = \nu = 0.625$  $\mu = \nu = 0.750$  $\mu = \nu = 0.875$  $\mu = \nu = 1.00$  Original image

Figure 9: Supplementing the original image information.

smaller $\mu$ and $\nu$ values exhibit similarities to those obtained with boundary control applied. Thus, by modifying the values of $\mu$ and $\nu$, we may be able to mitigate the feature loss and darkening issues caused by boundary control, which can be seen in Figure 9. What's more, our method ensures that noise levels meet the necessary threshold, but the information of images may exceed or fall short of the desired threshold because this is determined by $\alpha$ and $\beta$. Therefore, by adjusting the parameters $\mu$ and $\nu$, we can regulate the information of images, thereby improving the interpolation results.

## 5.5  FINAL RESULT

We collected images from the Internet and employed three different methods for image interpolation. Throughout the interpolation process, we maintained consistency in parameter settings, with detailed information available in the Appendix. From the interpolation results, we observe that our method effectively reduces artifacts and maximally preserves information compared to directly applying spherical linear interpolation. Furthermore, our approach outperforms methods involving noise introduction in preserving original image features, as illustrated in Figures 13 and 14.

## 6  CONCLUSION

In this paper, we propose a novel method that surpasses the limitations of spherical linear interpolation. Our approach establishes a unified framework for both spherical linear interpolation and directly introducing noise for interpolation methods, leveraging the strengths of each. Additionally, by imposing boundary control on noise and supplementing the original image information, our method effectively tackles the challenges posed by noise levels exceeding or falling below the denoising threshold. Through the correction of latent variables, our approach improves the interpolation results of natural images, achieving superior interpolation outcomes.

**Limitation and future work.** Our approach, like any method, is not without its drawbacks and constraints. Compared to directly introducing noise for interpolation, our method involves an extra step: mapping the images to the latent variables. This additional overhead will double the processing time. However, this extra overhead leads to better feature preservation. Furthermore, our paper mainly focuses on image data. Accordingly, its effectiveness in other modalities has not been validated, which is a potential limitation of our work. Thus, we will explore the possibility of our method in different modalities in our future work. We will also explore the possibility of applying our method to different scenarios, such as a) investigating the interpolation between natural and adversarial images (Zhang et al., 2022), b) studying the interpolation among different environments (Arjovsky et al., 2019), and c) exploring the interpolation between in-distribution and out-of-distribution data (Fang et al., 2022). Moreover, it is exciting to apply our method to many interesting scenarios, like interpolation between different person images, interpolation for low-level computer vision (Zamir et al., 2021), and interpolation for video generation (Liu et al., 2024).

# 7 ACKNOWLEDGMENTS

The work was supported by grants from the National Key R&D Program of China (No. 2021ZD0111801). YGZ and BH were supported by the NSFC General Program No. 62376235, Guangdong Basic and Applied Basic Research Foundation No. 2022A1515011652, HKBU Faculty Niche Research Areas No. RC-FNRA-IG/22-23/SCI/04, and HKBU CSD Departmental Incentive Scheme. TL is partially supported by the following Australian Research Council projects: FT220100318, DP220102121, LP220100527, LP220200949, IC190100031.

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

# A    PROOFS

## A.1    THE PROOF OF THEOREM 1

**Lemma 1.** *Let $\boldsymbol{X} = (X_1, ..., X_n) \in \mathbb{R}^n$ be a random vector with independent, sub-gaussian coordinates $X_i$ that satisfy $\mathbb{E}X_i^2 = 1$. Then*

$$\|\|\boldsymbol{X}\|_2 - \sqrt{n}\|_{\psi_2} \leq CK^2$$

*where $K = \max_i \|X_i\|_{\psi_2}$, $C$ is an absolute constant and we define:*

$$\|\boldsymbol{X}\|_{\psi_1} = \inf\{t > 0 : \mathbb{E}\exp(|\boldsymbol{X}|/t) \leq 2\}$$
$$\|\boldsymbol{X}\|_{\psi_2} = \inf\{t > 0 : \mathbb{E}\exp(\boldsymbol{X}^2/t^2) \leq 2\}$$

*Proof.* For simplicity, we assume that $K \geq 1$. We shall apply Bernstein's deviation inequality for the normalized sum of independent, mean zero random variables

$$\frac{1}{n}\|\boldsymbol{X}\|_2^2 - 1 = \frac{1}{n}\sum_{i=1}^{n}(X_i^2 - 1)$$

Since the random variable $X_i$ is sub-gaussian, $X_i^2 - 1$ is sub-exponential, and more precisely

$$\|X_i^2 - 1\|_{\psi_1} \leq C\|X_i^2\|_{\psi_1}$$
$$= C\|X_i\|_{\psi_2}^2$$
$$\leq CK^2$$

Applying Bernstein's inequality, we obtain for any $u \geq 0$ that

$$\mathbb{P}\{|\frac{1}{n}\|\boldsymbol{X}\|_2^2 - 1| \geq u\} \leq 2\exp(-\frac{cn}{K^4}\min(u^2, u))$$

This is a good concentration inequality for $\|X\|_2^2$, from which we are going to deduce a concentration inequality for $\|\boldsymbol{X}\|$. To make the link, we can use the following elementary observation that is valid for all numbers $z \geq 0$:

$$|z - 1| \geq \delta \text{ implies } |z^2 - 1| \geq \max(\delta, \delta^2)$$

We obtain for any $\delta \geq 0$ that

$$\mathbb{P}\{|\frac{1}{n}\|\boldsymbol{X}\|_2^2 - 1| \geq \delta\} \leq \mathbb{P}\{|\frac{1}{n}\|\boldsymbol{X}\|_2^2 - 1| \geq \max(\delta, \delta^2)\}$$
$$\leq 2\exp(-\frac{cn}{K^4} \cdot \delta^2) \quad (\text{for } u = \max(\delta, \delta^2))$$

Changing variables to $t = \delta\sqrt{n}$, we obtain the desired sub-gaussian tail

$$\mathbb{P}\{|\|\boldsymbol{X}\|_2 - \sqrt{n}| \geq t\} \leq 2\exp(-\frac{ct^2}{K^4}) \quad \text{for all } t \geq 0$$

As we know form Sub-gaussian properties, this is equivalent to the conclusion of the theorem. $\square$

**Theorem 1.** *The standard normal distribution $\mathcal{N}(\boldsymbol{0}, \boldsymbol{I}_n)$ in high dimensions is close to the uniform distribution on the sphere of radius $\sqrt{n}$.*

*Proof.* from Lemma 1, for the norm of $g \sim \mathcal{N}(0, \boldsymbol{I}_n)$ we have the following concentration inequality:

$$\mathbb{P}\{|\|g\|_2 - \sqrt{n}| \geq t\} \leq 2\exp(-ct^2) \quad \text{for all } t \geq 0$$

Let us represent $g \sim \mathcal{N}(0, \boldsymbol{I}_n)$ in polar form as

$$g = r\theta$$

where $r = \|g\|_2$ is the length and $\theta = g/\|g\|_2$ is the direction of $g$.

Concentration inequality says that $r = \|g\|_2 \approx \sqrt{n}$ with high probability, so

$$g \approx \sqrt{n}\theta \sim \text{Unif}(\sqrt{n}S^{n-1})$$

In other words, the standard normal distribution in high dimensions is close to the uniform distribution on the sphere of radius $\sqrt{n}$, i.e.

$$\mathcal{N}(0, \boldsymbol{I}_n) \approx \sqrt{n}\theta \sim \text{Unif}(\sqrt{n}S^{n-1})$$

$\square$

## A.2 THE PROOF OF THEOREM 2

**Definition 1.** *A random vector $X$ in $\mathbb{R}^n$ is called **isotropic** if*

$$\sum(X) = \mathbb{E}XX^T = I_n$$

*where $I_n$ denotes the identity matrix in $\mathbb{R}^n$.*

**Lemma 2.** *A random vector $X$ in $\mathbb{R}^n$ is isotropic if and only if*

$$\mathbb{E}\langle X, x\rangle^2 = \|x\|_2^2 \quad \text{for all } x \in \mathbb{R}^n$$

*Proof.* Recall that two symmetric $n \times n$ matrices $A$ and $B$ are equal if and only if $x^T A x = x^T B x$ for all $x \in \mathbb{R}^n$. Thus $X$ is isotropic if and only if

$$x^T(\mathbb{E}XX^T)x = x^T I_n x \quad \text{for all } x \in \mathbb{R}^n$$

The left side of this identity equals $\mathbb{E}\langle X, x\rangle^2$, and the right side is $\|x\|_2^2$. $\qquad\square$

**Lemma 3.** *Let $X$ be an isotropic random vector in $\mathbb{R}^n$. Then*

$$\mathbb{E}\|X\|_2^2 = n$$

*Moreover, if $X$ and $Y$ are two independent isotropic random vectors in $\mathbb{R}^n$, then*

$$\mathbb{E}\langle X, Y\rangle^2 = n$$

*Proof.* To prove the first part, we have

$$\begin{aligned}
\mathbb{E}\|X\|_2^2 = \mathbb{E}X^T X = \mathbb{E}\operatorname{tr}(X^T X) \quad &\text{(viewing } X^T X \text{ as a } 1 \times 1 \text{ matrix)} \\
= \mathbb{E}\operatorname{tr}(XX^T) \quad &\text{(by the cyclic property of trace)} \\
= \operatorname{tr}\mathbb{E}(XX^T) \quad &\text{(by linearity)} \\
= \operatorname{tr}(I_n) \quad &\text{(by isotropy)} \\
= n
\end{aligned}$$

To prove the second part, we use a conditioning argument. Fix a realization of $Y$ and take the conditional expectation (with respect to $X$) which we denote $\mathbb{E}_X$. The law of total expectation says that

$$\mathbb{E}\langle X, Y\rangle^2 = \mathbb{E}_Y \mathbb{E}_X[\langle X, Y\rangle^2 | Y],$$

where by $\mathbb{E}_Y$ we of course denote the expectation with respect to $Y$. To compute the inner expectation, we apply Lemma 2. with $x = Y$ and conclude that the inner expectation equals $\|Y\|_2^2$. Thus

$$\begin{aligned}
\mathbb{E}\langle X, Y\rangle^2 &= \mathbb{E}_Y \|Y\|_2^2 \\
&= n \quad \text{(by the first part of lemma)}
\end{aligned}$$

$\qquad\square$

**Theorem 2.** *In high-dimensional spaces, independent and isotropic random vectors tend to be almost orthogonal*

*Proof.* Let us normalize the random vectors X and Y in Lemma 3 setting

$$\bar{X} := \frac{X}{\|X\|_2} \quad \text{and} \quad \bar{Y} := \frac{Y}{\|Y\|_2}$$

Lemma 3 is basically telling us that $\|X\|_2 \asymp \sqrt{n}$, $\|Y\|_2 \asymp \sqrt{n}$ and $\langle X, Y\rangle \asymp \sqrt{n}$ with high probability, which implies that

$$|\langle \bar{X}, \bar{Y}\rangle| \asymp \frac{1}{\sqrt{n}}$$

Thus, in high-dimensional spaces independent and isotropic random vectors tend to be almost orthogonal. $\qquad\square$

## B  Experiments with Models Training in Single Domain

**Parameter choices** To facilitate comparison with spherical linear interpolation results, we maintain the condition $\alpha/\beta = \sin\left(\frac{\pi}{2} \cdot \lambda\right) / \cos\left(\frac{\pi}{2} \cdot \lambda\right)$, and ensure that $\sqrt{\alpha^2 + \beta^2 + \gamma^2} = 1$ when computing $\alpha$ and $\beta$. And we set $\mu = 2.0 * \alpha/(\alpha + \beta)$, $\nu = 2.0 * \beta/(\alpha + \beta)$. Additionally, several other parameters, albeit hyperparameters, have predefined ranges for user convenience. For instance, the boundary ranges from 2.0 to 2.4, $\gamma \in [0, \sqrt{0.1}]$. Users only need to determine the value of $\lambda$ to specify the style of interpolation results.

### B.1  The impact of the boundary

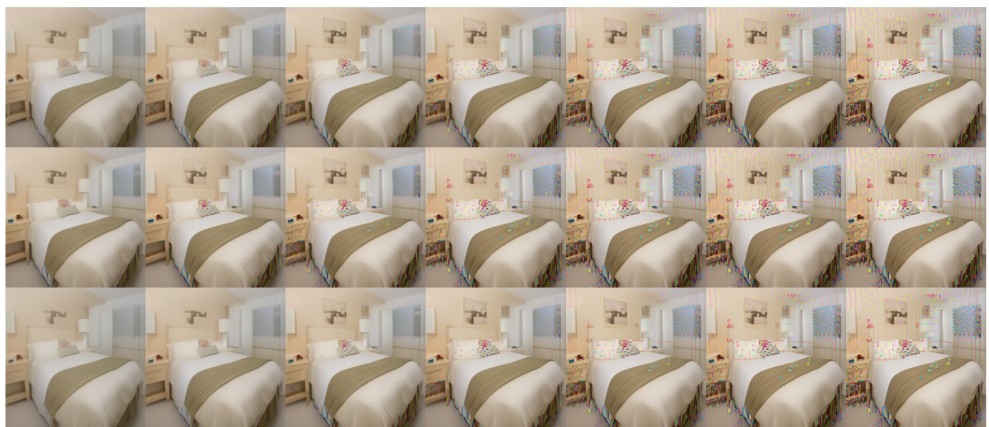

Boundary=2.0  Boundary=2.2  Boundary=2.4  Boundary=2.6  Boundary=2.8  Boundary=3.0  Boundary=3.2

Figure 10: The impact of the boundary. From top to bottom: controlling noise before interpolation, controlling noise after interpolation, and controlling noise both before and after interpolation. From left to right: the coefficient ratio of the noise boundary to the variance are $[2.0, 2.2, 2.4, 2.6, 2.8, 3.0, 3.2]$.

We compared three boundary control methods: control before interpolation, control after interpolation, and control before and after interpolation, as shown in Figure 10. From the figure, we can observe that all three methods introduced a similar level of blurriness, indicating a loss of image information, and applying constraints to noise both before and after interpolation is more effective in reducing artifacts.

### B.2  Interpolation of images with models trained on lsun bedroom-256

We searched online for images of bedroom and used a diffusion model trained exclusively on LSUN Bedroom-256 images for interpolation. We gradually increased the value of $\lambda$ to modify the style of the interpolation images and ensuring that other parameters are within the specified range. The results are shown in Figure11.

### B.3  Interpolation of images with models trained on lsun cat-256

We searched online for images of cat and used a diffusion model trained exclusively on LSUN Cat-256 images for interpolation. We gradually increased the value of $\lambda$ to modify the style of the interpolation images and ensuring that other parameters are within the specified range. The results are shown in Figure12.

### B.4  Comparison of results from different methods

We compared our method with spherical linear interpolation and the method of introducing noise for interpolation, using models separately trained on LSUN Cat-256 and LSUN Bedroom-256 datasets.

Input                    Generated Interpolations                    Input

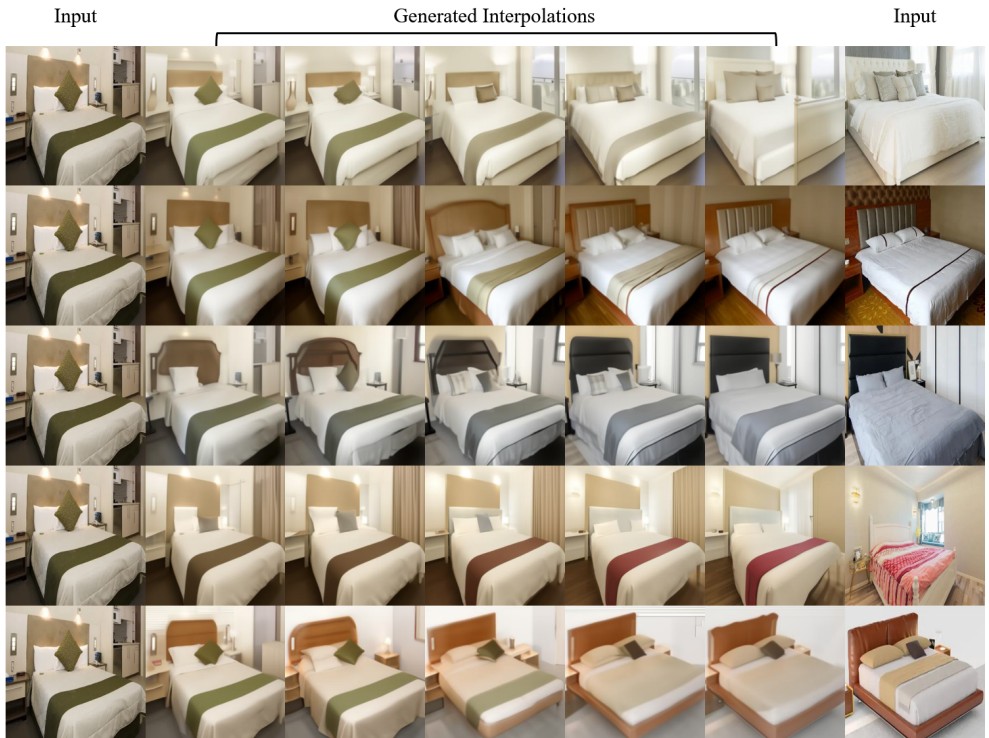

Figure 11: Interpolation with natural images. By modifying $\lambda$, our method can generate interpolated results with different image styles.

The results are displayed in Figure 13 and Figure 14. From the figures, it's clear that spherical linear interpolation introduces significant artifacts, while introducing noise for interpolation introduces extra information. In contrast, our method not only preserves the original image informations but also enhances the quality of images.

## C  EXPERIMENTS ON STABLE DIFFUSION

### C.1  STABLE DIFFUSION

We extended our experiments on Stable Diffusion and compared it with other methods. Due to the differences in the form of $\boldsymbol{\mu}(\boldsymbol{x}_t, t)$ and $\sigma(t)$ in Stable Diffusion, there have been significant changes in its latent variables. However, the challenges faced by different interpolation methods are similar: spherical linear interpolation produces images with noticeable defects (Figure 21 - Figure 25), while the method of introducing noise for interpolation introduces additional information (Figure 16 - Figure 20). Due to the highly unstructured latent space of the Stable Diffusion, it becomes challenging to interpolate between two image samples, as depicted in Figure 15.Therefore, we consider interpolating latent variables in the noisy image space, here we chose to interpolate the images when $t = 700$.

### C.2  EXPERIMENTAL RESULTS

We collected various images online to interpolate on Stable Diffusion. The results are shown below. To facilitate comparison with spherical linear interpolation results, we maintain the condition $\alpha/\beta = \sin\left(\frac{\pi}{2} \cdot \lambda\right) / \cos\left(\frac{\pi}{2} \cdot \lambda\right)$, and ensure that $\sqrt{\alpha^2 + \beta^2 + \gamma^2} = 1$ when computing $\alpha$ and $\beta$. Additionally, the boundary is set to 2.0, $\gamma \in [0, \sqrt{0.1}]$, and $\mu = 1.2 * \alpha/(\alpha + \beta)$, $\nu = 1.2 * \beta/(\alpha + \beta)$. Users need to determine the value of $\lambda$ to modify the style of interpolation results.

Input      Generated Interpolations      Input

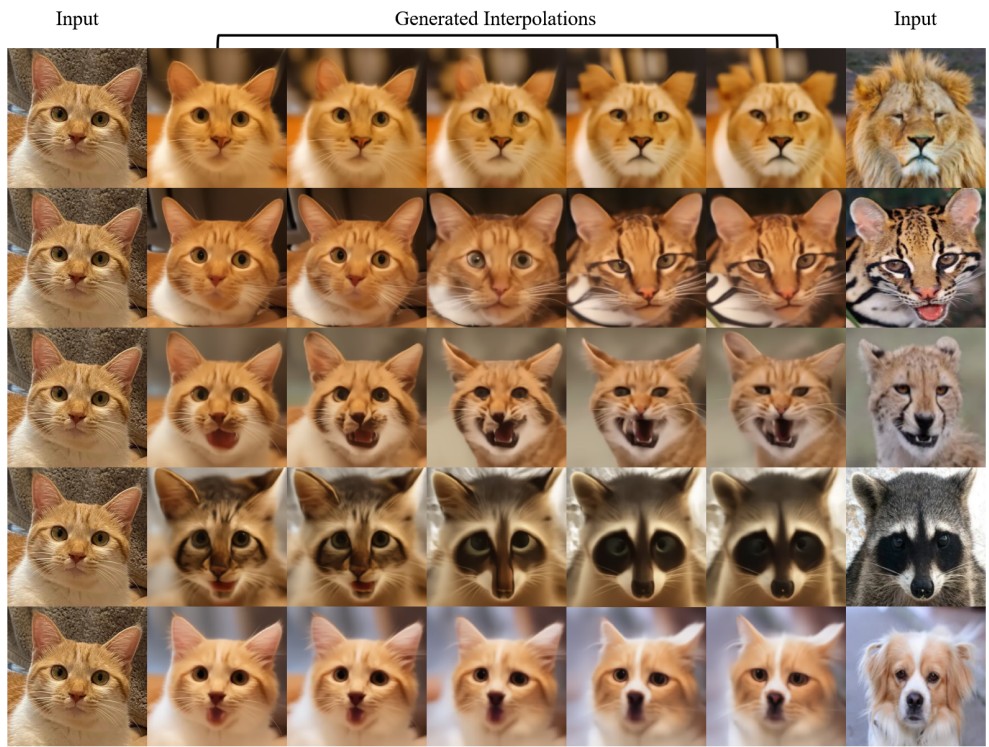

Figure 12: Interpolation with natural images. By modifying $\lambda$, our method can generate interpolated results with different image styles.

Input

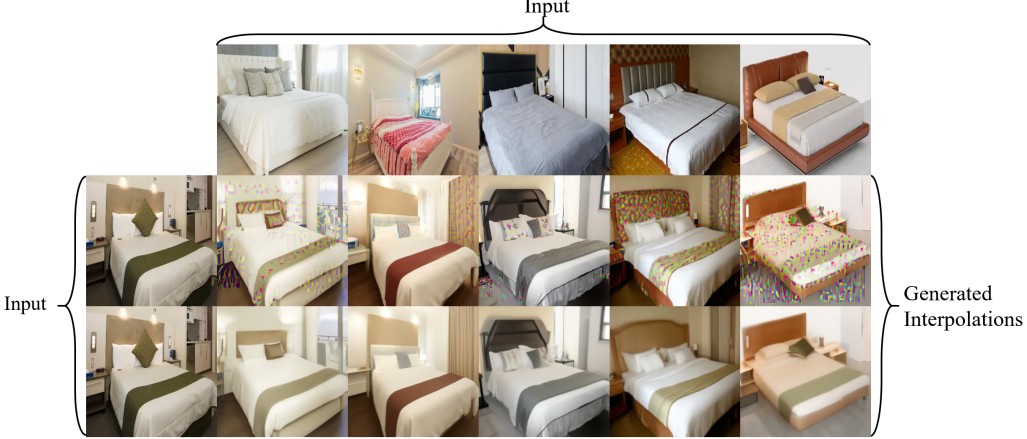

Input      Generated Interpolations

Figure 13: Comparison between spherical linear interpolation and our method. The top and left-most images represent the original images. The second row displays the results of spherical linear interpolation, while the third row shows the outcomes of our method.

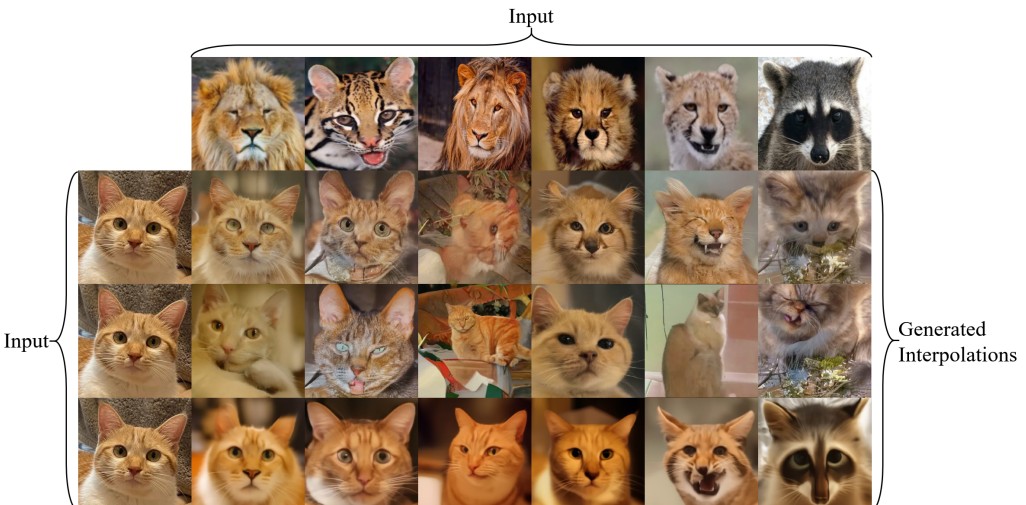

Figure 14: Comparison between the method of introducing noise for interpolation and our method. The top and leftmost images show the original images. The second and third rows display the interpolation results obtained by directly introducing noise. The fourth row illustrates the outcomes of our method.

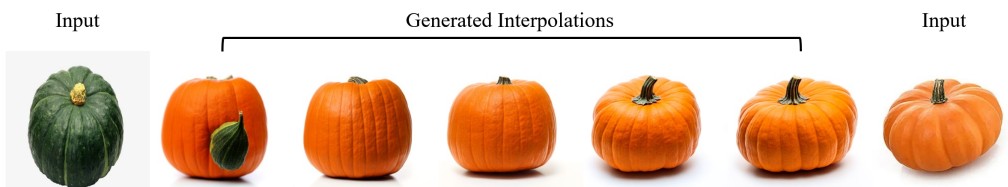

Figure 15: Spherical linear interpolation results when the images are encoded into the noise space.

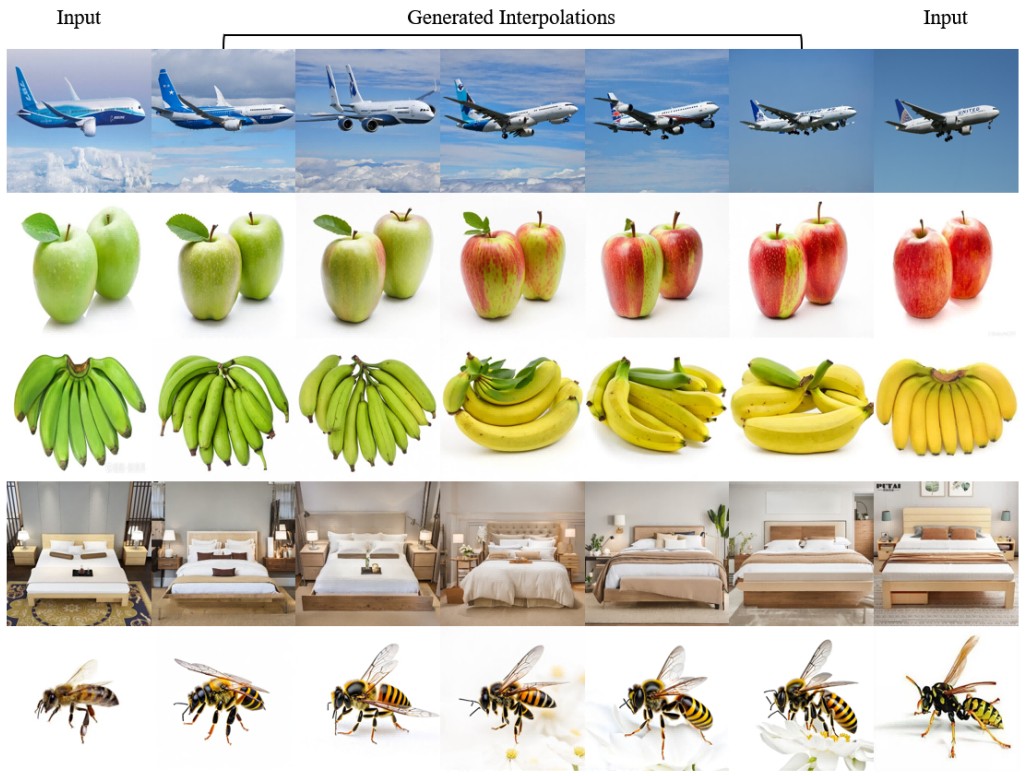

Figure 16: Interpolation results with Stable Diffusion (Introducing Noise) (1/5).

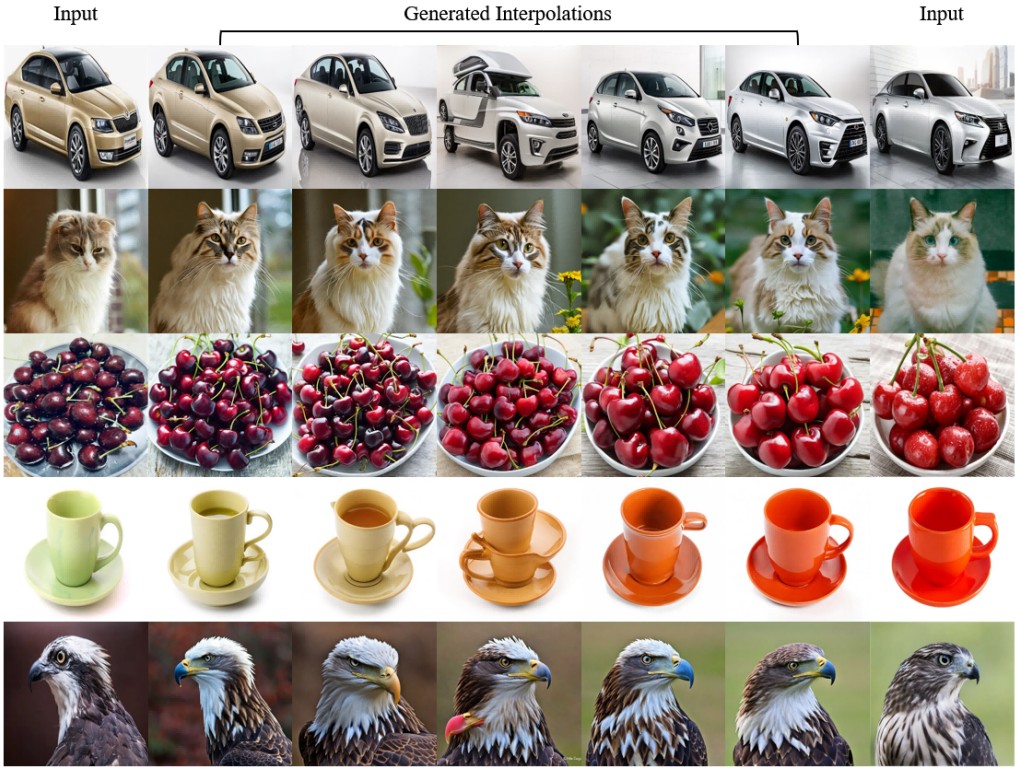

Figure 17: Interpolation results with Stable Diffusion (Introducing Noise) (2/5).

Input        Generated Interpolations        Input

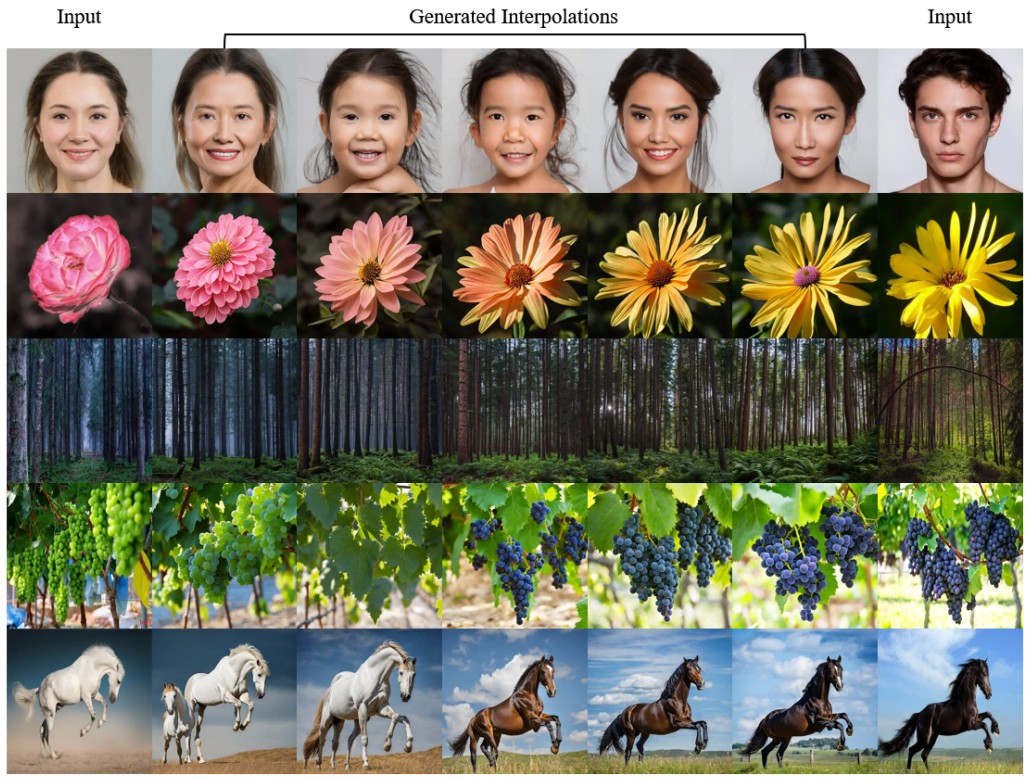

Figure 18: Interpolation results with Stable Diffusion (Introducing Noise) (3/5).

Input        Generated Interpolations        Input

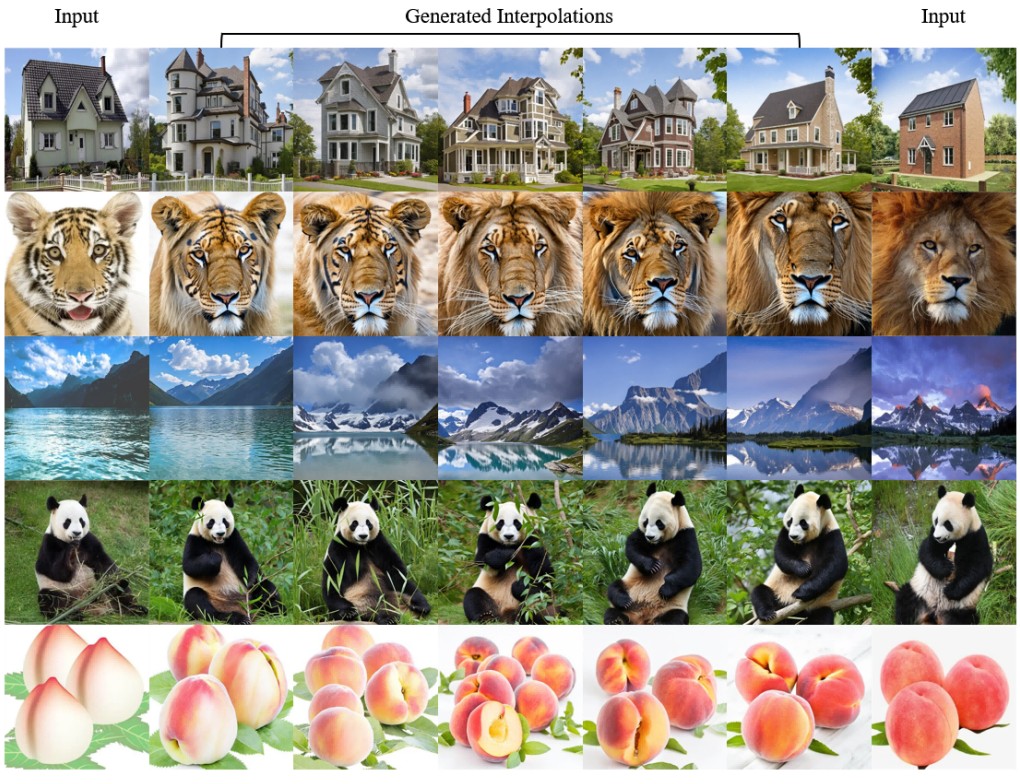

Figure 19: Interpolation results with Stable Diffusion (Introducing Noise) (4/5).

Input           Generated Interpolations           Input

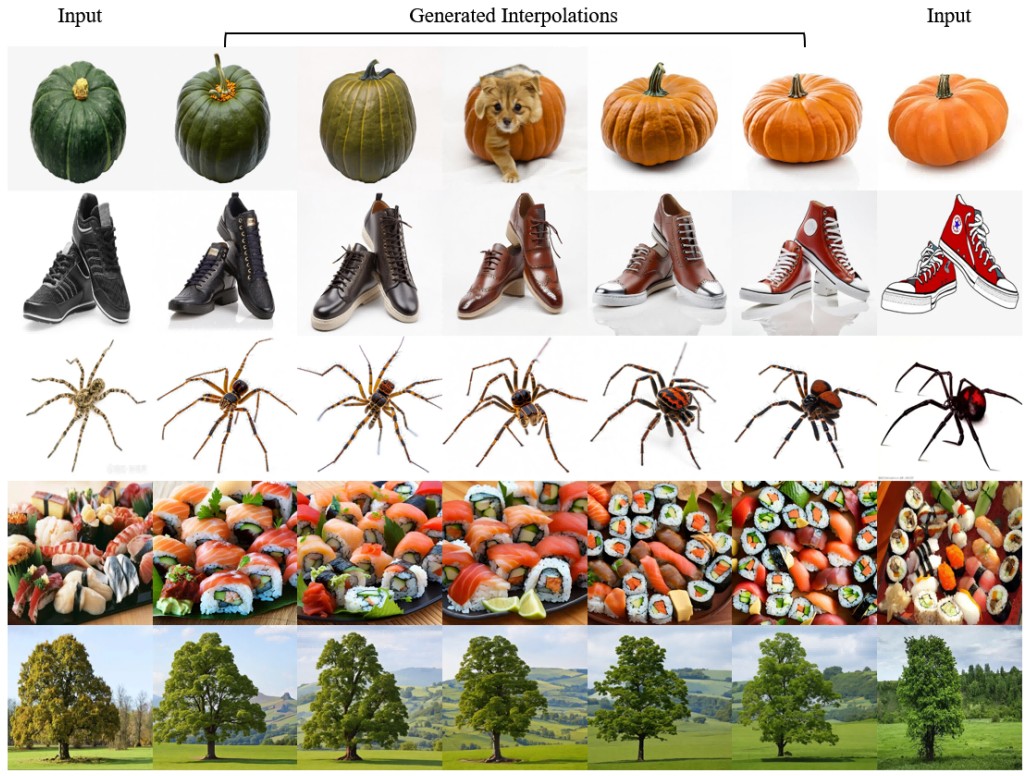

Figure 20: Interpolation results with Stable Diffusion (Introducing Noise) (5/5).

Input           Generated Interpolations           Input

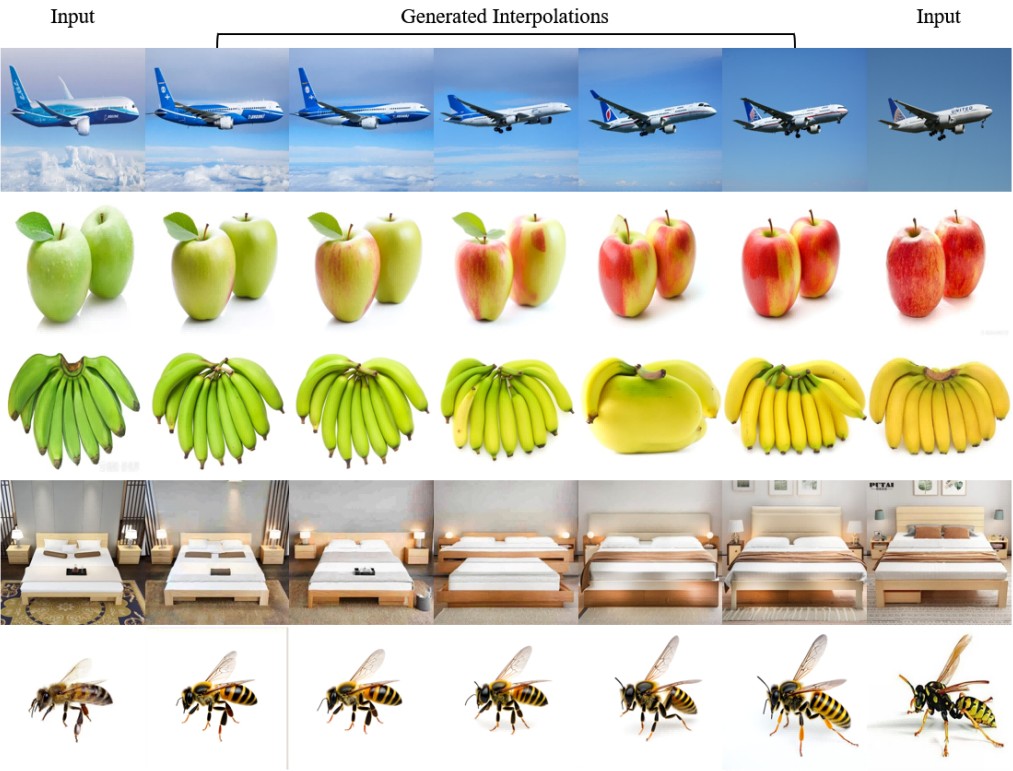

Figure 21: Interpolation results with Stable Diffusion (Spherical Linear Interpolation) (1/5).

Input              Generated Interpolations              Input

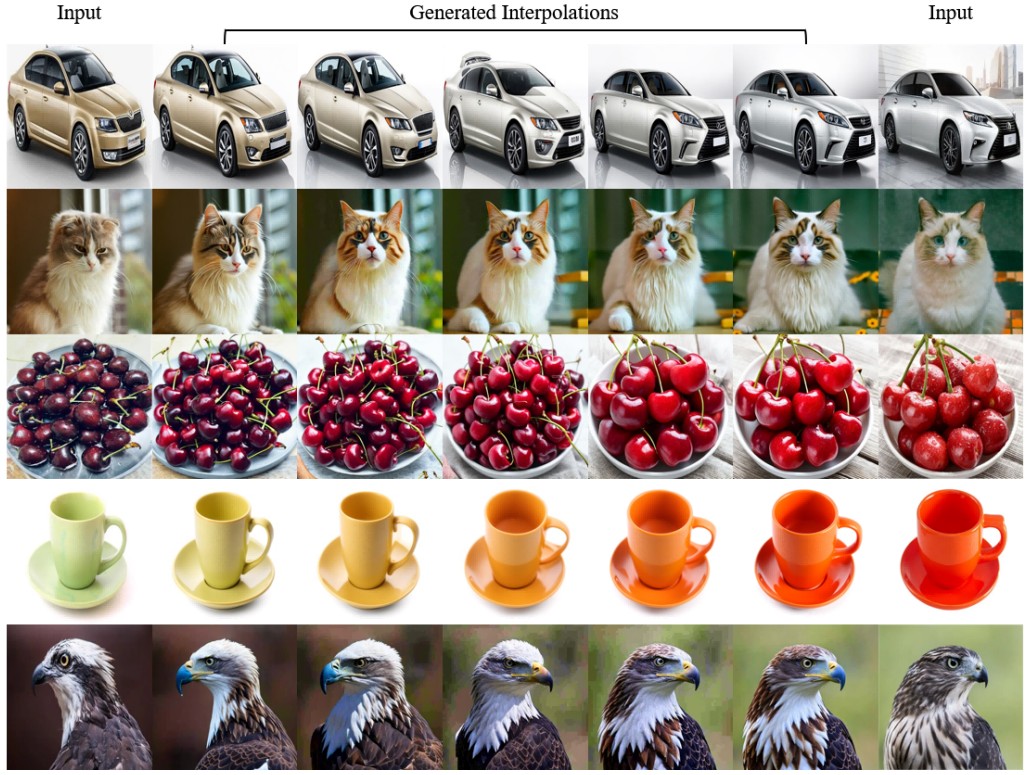

Figure 22: Interpolation results with Stable Diffusion (Spherical Linear Interpolation) (2/5).

Input              Generated Interpolations              Input

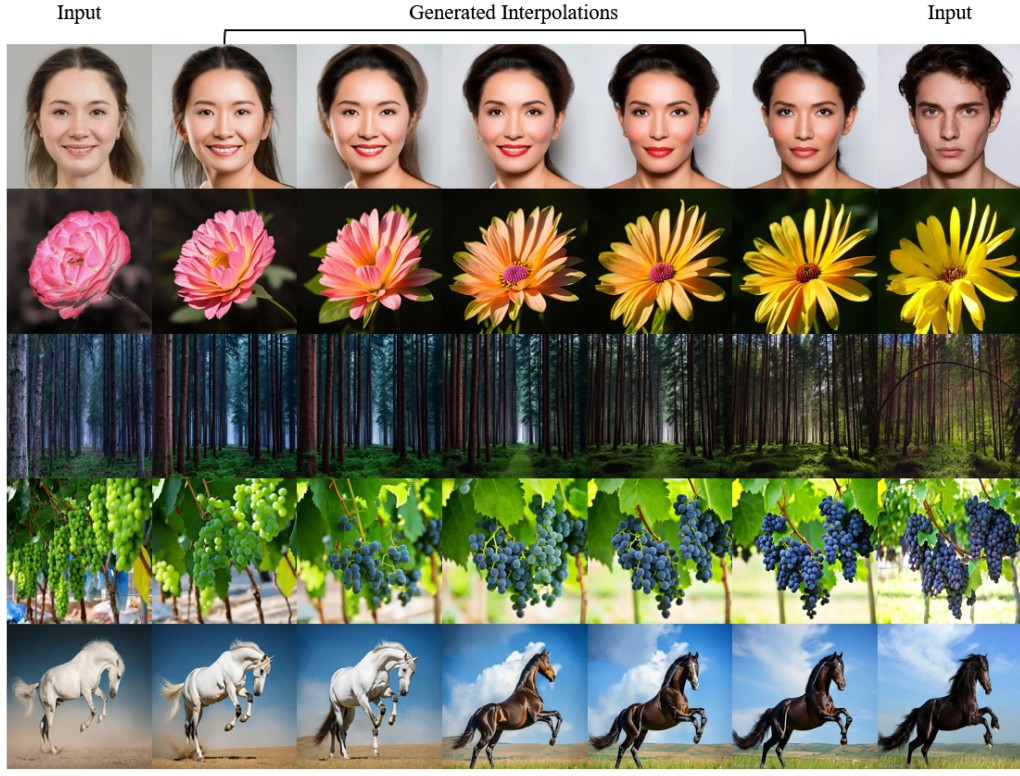

Figure 23: Interpolation results with Stable Diffusion (Spherical Linear Interpolation) (3/5).

Input           Generated Interpolations           Input

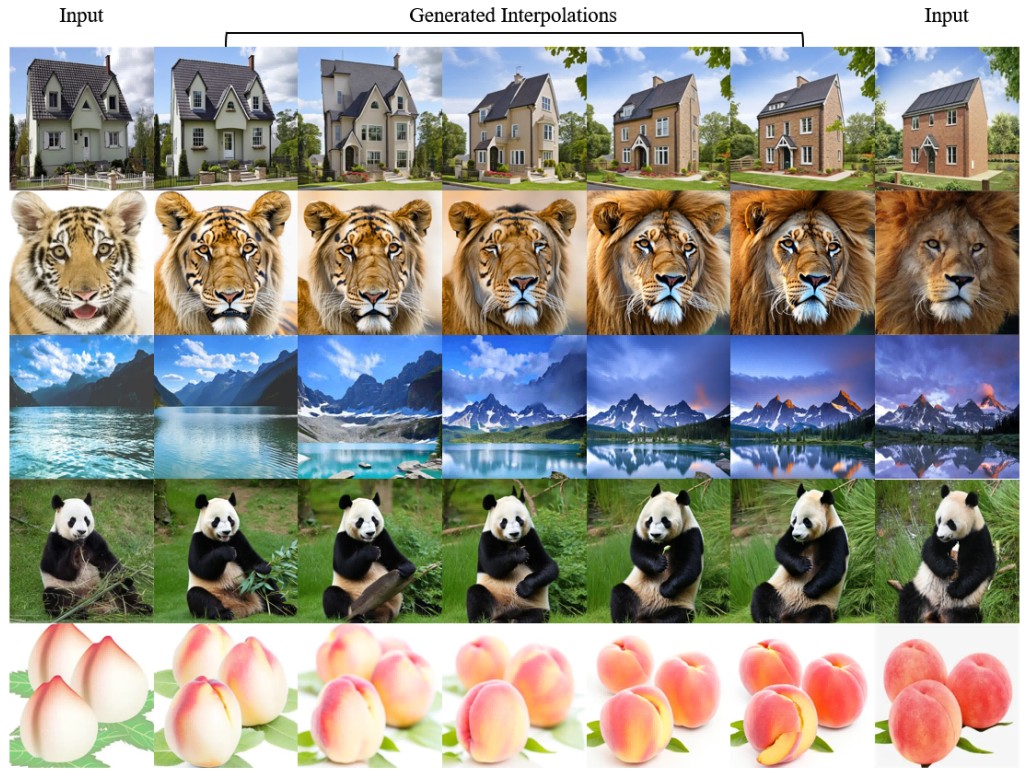

Figure 24: Interpolation results with Stable Diffusion (Spherical Linear Interpolation) (4/5).

Input           Generated Interpolations           Input

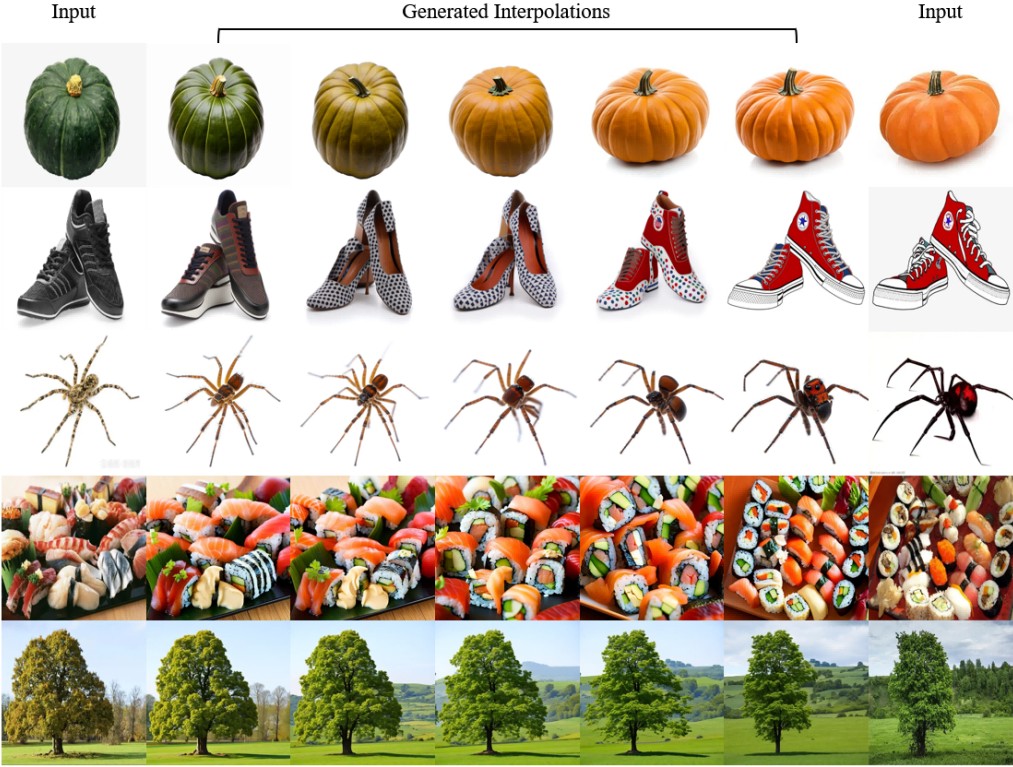

Figure 25: Interpolation results with Stable Diffusion (Spherical Linear Interpolation) (5/5).

Input · Generated Interpolations · Input

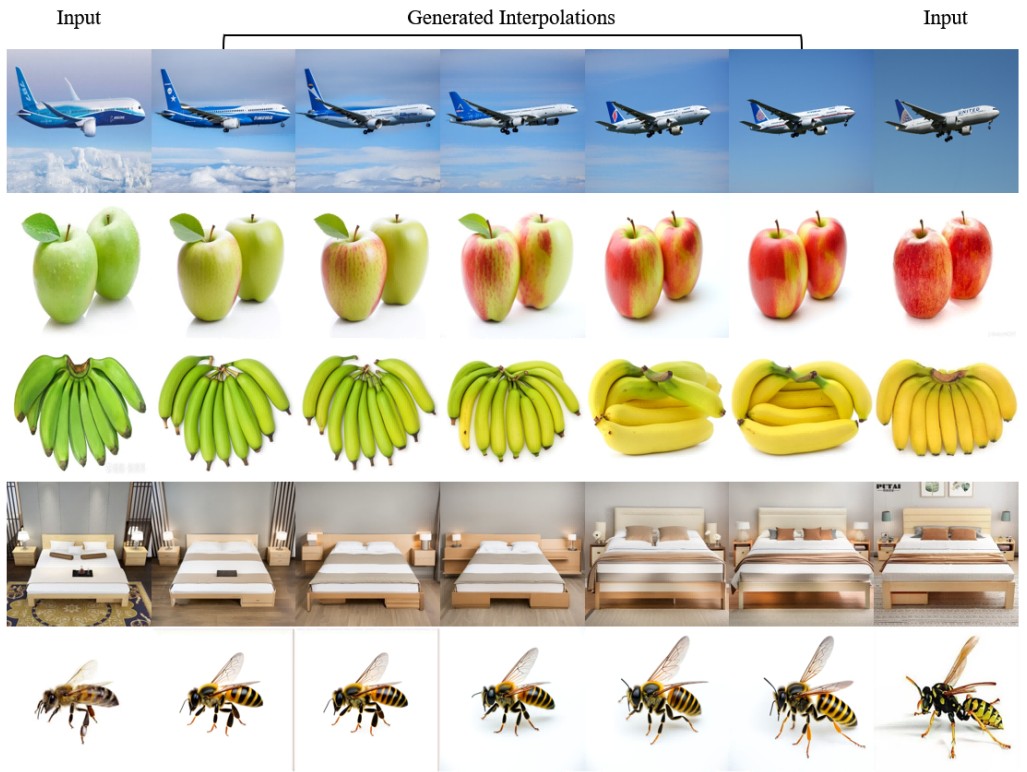

Figure 26: Interpolation results with Stable Diffusion (NoiseDiffusion) (1/5) .

Input · Generated Interpolations · Input

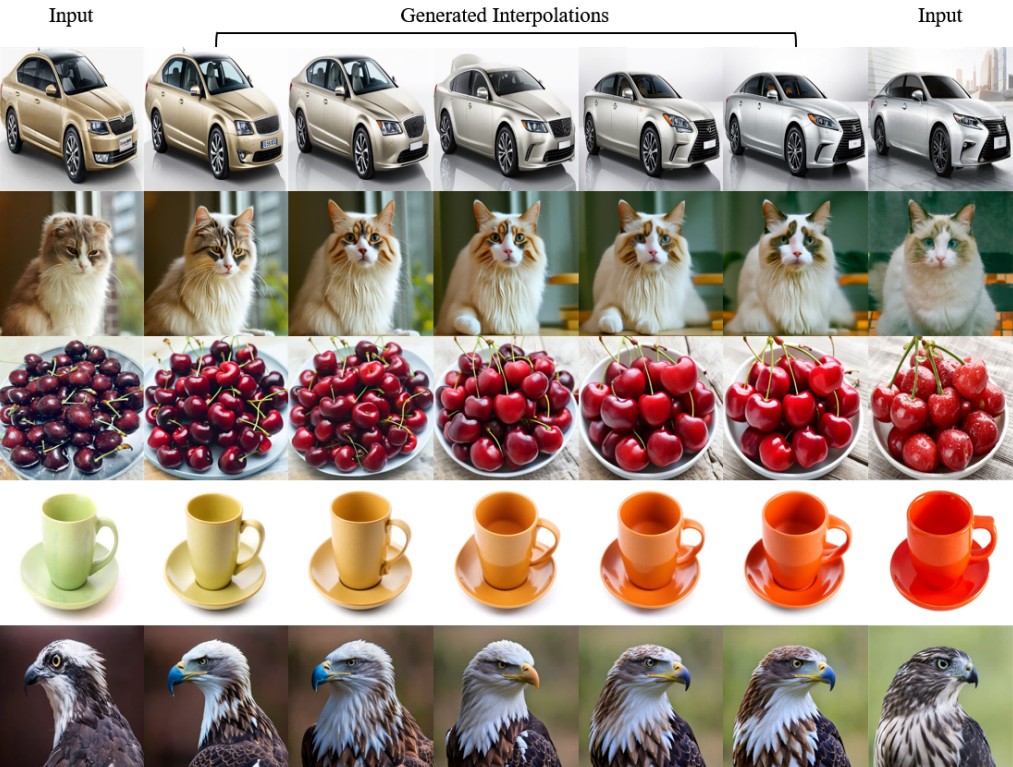

Figure 27: Interpolation results with Stable Diffusion (NoiseDiffusion) (2/5) .

Input                    Generated Interpolations                    Input

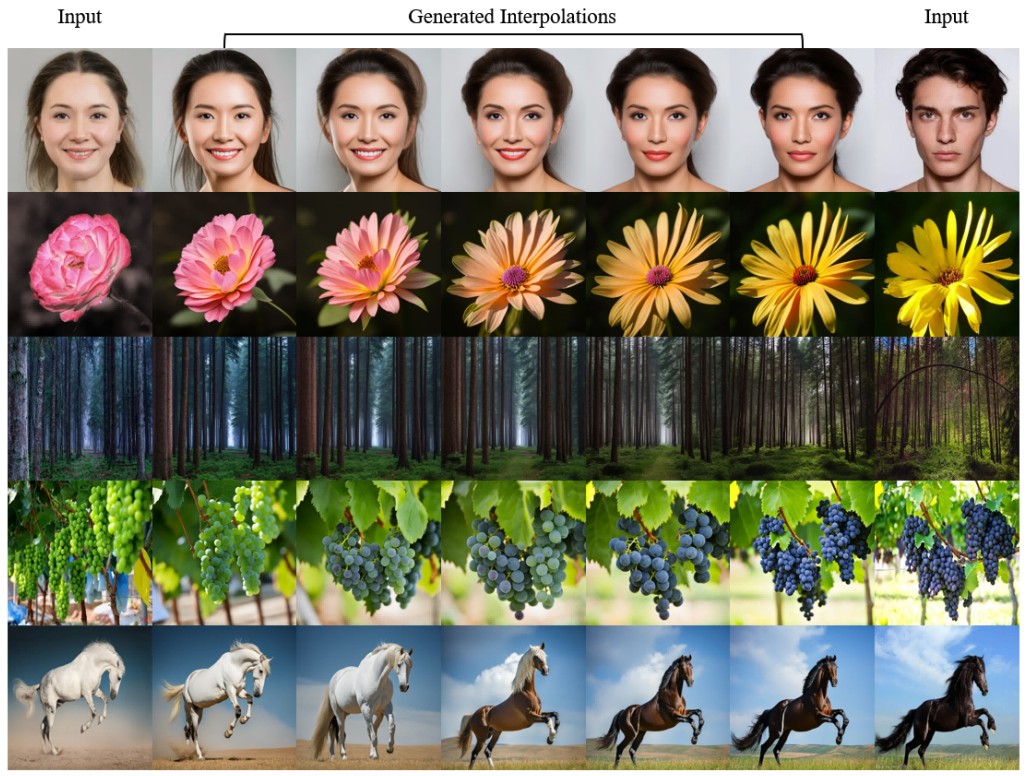

Figure 28: Interpolation results with Stable Diffusion (NoiseDiffusion) (3/5) .

Input                    Generated Interpolations                    Input

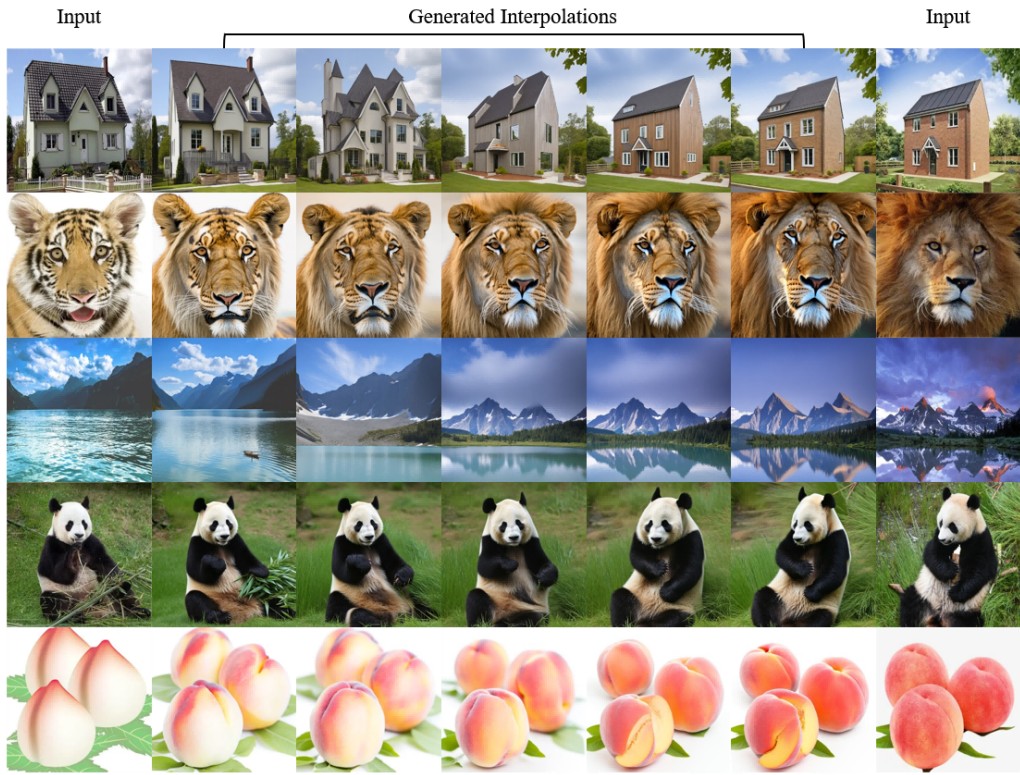

Figure 29: Interpolation results with Stable Diffusion (NoiseDiffusion) (4/5) .

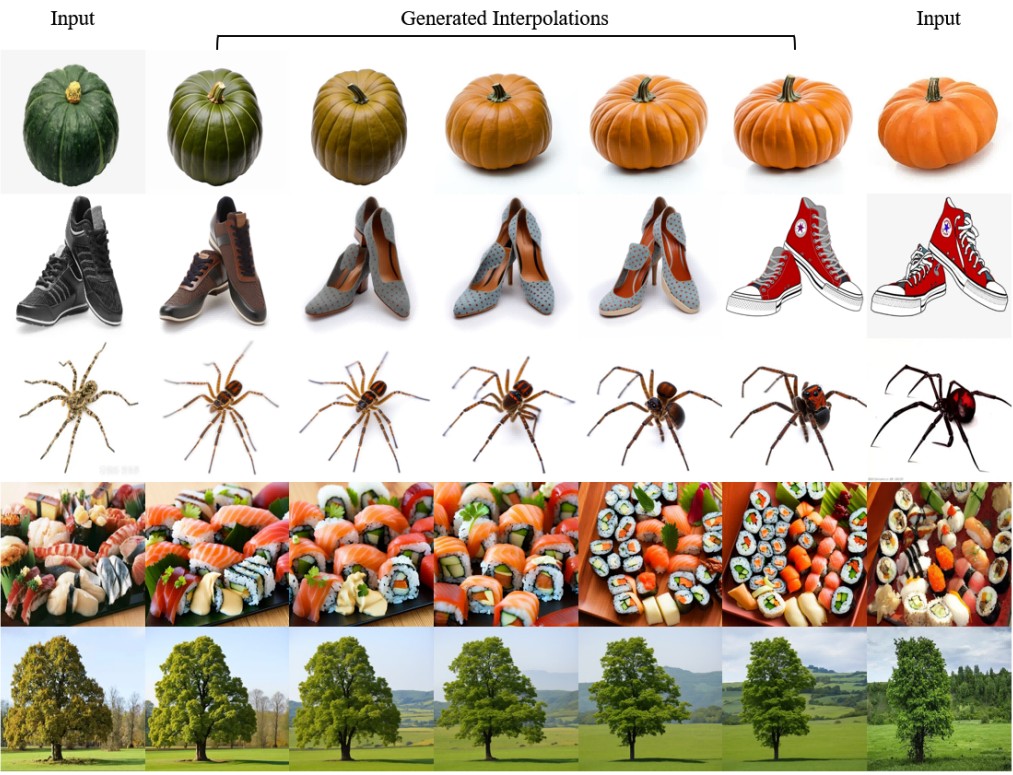

Figure 30: Interpolation results with Stable Diffusion (NoiseDiffusion) (5/5).

