# OpenReview forum: "NoiseDiffusion: Correcting Noise for Image  Interpolation  with Diffusion Models beyond Spherical Linear Interpolation"
_ICLR.cc/2024/Conference — ICLR 2024 spotlight_

### Official Review · Reviewer_A19z · 2023-10-30

**Soundness:** 4 excellent
**Presentation:** 4 excellent
**Contribution:** 4 excellent
**Rating:** 8
**Confidence:** 4

**Summary:**

A new approach to image interpolation with diffusion models is presented, where a new image is interpolated from two other images (in contrast to interpolating pixels within a single image).  While exisitng methods work reasonably well with images generated by a diffusion model, they do not work well with natural images not generated from the diffusion model.  Resulting interpolated images exhibit considerable artefacts.  Theory is introduced to explain the reason for the failure and suggest how to mitigate it.  A novel approach is presented to add appropriate noise to the natual images considered but at the same time ensure the interpolated image is faithful to the two images it is generated from.  A number of experiments are presented that demonstrate the improvement in interpolated image quality compared to the standard spherical interpolation approach.

**Strengths:**

The paper is clear and well written.  The background and existing approaches are clearly explained, highlighting the tradeoff between realsim (i.e. avoiding artefacts) and faithfulness (i.e. ensuring consistency with the source images).

The reason for the existing failure for natural images is related the mismatched noise levels, which arises due to the concentration of probability in high dimensions.  A numerical experiment (Fig. 3) is performed to support this hypothesis and Theorem 1 is presented to provide further support.  I believe Theorem 1 is a fairly well-known result.  Nevertheless, a proof is presented in Appendix A (I initially did not appreciate a proof was provided; in the main body of the article the reader should be pointed to Appendix A for the proof).

Further theory is presented in Theorem 2 to guide a solution to the problem (again, in the main body the reader should be pointed to the proof of Theorem 2 in Appendix B).  Based on the provided theory a novel diffusion-based image interpolation approach is presented, combining an SDEdit-like stage, followed by interpolation in latent space, followed by denoising with the diffusion model.  The interpolation approach introduces a number of free parameters to address contrast issues and trade-off consistence with each source image.  Standard spherical interpolation and linear interpolation combined with SDEdit are recovered as special cases for specific parameter choices.

The resulting interpolation method is simple but appears to work well.  I expect it to be of considerable practical use.

**Weaknesses:**

While a number of supporting experiments are presented, these could be more compelling.  Experiments are performed to demonstrate the impact of different parameter choices, which are helpful.  In terms of overall interpolation quality in comparison to existing techniuqes, a number of examples a presented where the proposed method is demonstrated to be visually superior to existing approaches.  While further examples are presented in the appendices, in total just a handful of images are considered.  Given the artefacts that arise with existing techniques, I don't doubt that the proposed method is superior.  However, to further support this it would be useful to have some metric to summarize performance over a large test data-set.  I appreciate this may not be trivial but I wonder if the authors have any thoughts on how they could assess performance over a large test-set?

Boundary control methods are introduce in Section 4.4 but are not explained.  The appendix is referenced but that only contains an additional figure of results.  The boundry control methods should be discussed somewhere.  In general Appendix B feel incomplete.  While I appreciate papers should stand on their own, without Appendices, if that is the case then further detail needs to be provided in the main body.

There seem to be a number of minor typos throughout where spaces a missing, typically following a period or comma.

**Questions:**

Could the authors propose a summary metric to qualtify the performance of their approach over a large test-data set?  At present results are presented for just a handful of test images.  While is seems very likely the improved performance of the proposed method will generalize to other data it would be more compelling if the authors could demonstrate this quantitatively.

---

> ### Author Response · Authors · 2023-11-18
>
> Thank you for your valuable comments and constructive feedback on our manuscript. We appreciate the time and effort you have dedicated to reviewing our work. In response to your insightful suggestions, we have made several revisions to address the key concerns raised during the review process.
>
> **Q.1**: Propose a metric to quantify the performance on a large test dataset.
> > However, to further support this it would be useful to have some metric to summarize performance over a large test data-set.I appreciate this may not be trivial but I wonder if the authors have any thoughts on how they could assess performance over a large test-set?
> Could the authors propose a summary metric to qualtify the performance of their approach over a large test-data set? At present results are presented for just a handful of test images. While is seems very likely the improved performance of the proposed method will generalize to other data it would be more compelling if the authors could demonstrate this quantitatively.
> >
> ***Ans for Q.1***:
>
> Thank you for your constructive suggestions. Accordingly, we have conducted more experiments using difference models and datasets. These results are reported and discussed in our revision.
>
>
> - We have conducted experiments on Stable Diffusion model with datasets like  eagle, lion-tiger, peach and so on. The high quality results are reported in the Appendix C.2, which shows that our method can consistently achieve good performance under different settings.Specifically, we have tested not only the scenarios where the drift coefficient of the model is zero and non-zero but also tested whether the model's variance explodes, which was done to determine whether the diffusion process belongs to a Variance Exploding SDE(VE-SDE) or a Variance Preserving SDE(VP-SDE)[1]. This indicates that our approach may be useful in almost all diffusion models.
>
> -  We agree that an appropreate quantitative metric is crucial for compare difference methods. However,  as you mentioned, designing such a metric is challenging  since it should not only be based on pixels but also consider the semantic aspect to assess the quality of the interpolated images. The challenge may be the reason why previous methods in the literature did not involve such a metric. In this context, we would like to enhance the persuasiveness of our results by reporting additional results in the appendix. We will leave it as our future work to explore methods for quantifying the quality of interpolated images.
>
> **Q.2**: Boundary control are not explained.
> >Boundary control methods are introduce in Section 4.4 but are not explained. The appendix is referenced but that only contains an additional figure of results. The boundry control methods should be discussed somewhere. In general Appendix B feel incomplete.
>
> ***Ans for Q.2***:
>
> We apologize for missing the introduction of the boudary control. Following your valuable suggestion, we have highlighted the definition and explanation of boudary control in our revision of Chapter 4.5. Specifically,combining the "68–95–99.7 rule" and considering our analysis of how noise above the denoising threshold impacts images, data points exhibiting significant deviations from the mean are considered potential sources of image artifacts.Therefore, we control the boundary of noise.
>
>
>
> **Q.3**: Writing problem.
> >There seem to be a number of minor typos throughout where spaces a missing, typically following a period or comma.
> >
> ***Ans for Q.3***:
>
> Thanks for your kind suggestion. In respect to your careful reviewing, we have re-examined the entire paper and fixed all typos. Meanwhile, following your advice, we have added the citation of our theorem to the main text.
>
> [1]Song, Y., Sohl-Dickstein, J., Kingma, D. P., Kumar, A., Ermon, S., and Poole, B. Score-based generative modeling through stochastic differential equations. In International Conference on Learning Representations, 2021.

---

> ### Author Response · Authors · 2023-11-20
> **Welcome for more discussions**
>
> Thanks for your valuable time in reviewing and insightful comments. Following your comments, we have tried our best to provide responses and revise our paper. Here is a **summary of our response** for your convenience:
> - (1) **Metric to quantify the performance on a large test dataset**: As you mentioned, designing such a metric is challenging since it should not only be based on pixels but also consider the semantic aspect to assess the quality of the interpolated images. We would like to enhance the persuasiveness of our results by reporting additional results in the appendix. And We will leave it as our future work to explore methods for quantifying the quality of interpolated images.
> - (2) **Boundary control**: We highlighted the definition and explanation of boudary control in our revision of Chapter 4.5.
> - (3) **Typo error**: We have re-examined the entire paper and fixed all typos and added the citation of our theorem to the main text.
>
>
> We humbly hope our repsonse has addressed your concerns. If you have any additional concerns or comments that we may have missed in our responses, we would be most grateful for any further feedback from you to help us further enhance our work.

---

> > ### Comment · Reviewer_A19z · 2023-11-23
> >
> > Thanks for the response and revisions to the manuscript.  My original accept recommendation stands.

---

### Official Review · Reviewer_YU98 · 2023-11-06

**Soundness:** 3 good
**Presentation:** 3 good
**Contribution:** 3 good
**Rating:** 8
**Confidence:** 3

**Summary:**

The paper explores and discusses the limitations of linear spherical interpolation in the context of image interpolation, particularly when applied to images not generated by diffusion models. It investigates how inappropriate noise introduction and estimation (overestimation and underestimation of noise level) affects image quality, leading to artifacts and loss of original style in the interpolated images. The study aims to address these limitations by proposing a novel interpolation approach that strategically imposes boundary constraints (i.e. clipping) on the introduced noise (at inversion steps), aligning with the hypersphere centered around the Gaussian distribution. This approach aims to mitigate noise artifacts and improve image quality, preserving the essential characteristics of the original image while efficiently removing artifacts. Additionally, the paper explores the integration of different interpolation methods, such as linear spherical interpolation and SDEdit, to overcome the identified limitations and achieve satisfactory results in both training and natural image domains.

**Strengths:**

• Identifies and addresses limitations in current linear spherical interpolation techniques, particularly in handling natural images (not generated by diffusion models).

    • Offers an approach to enhance image interpolation quality and address noise-related issues (i.e. noise artifacts and loss of style dues to a mismatch between the actual noise strength and its estimation) through and imposing a clipping constraint on the noise components.

**Weaknesses:**

INITIAL REVIEW:

• The paper's contribution heavily relies on specific parameter selection (e.g., parameters in eq.(14), boundary values in section 3.5, and the interpolation coefficient $\lambda$ in slerp), yet practical guidance for selecting these parameters remains unclear.

    • Computational overhead and potential limitations of implementing the proposed method in practical scenarios lack detailed explanation. The experiments are conducted under a simplified scenario where the drift coefficient $\mu(x_t,t)$ is set to zero.

    • Further exploration is needed to assess the level of improvement and applicability in diverse domains or image types, as seen in Figure 6.

    • The paper lacks explicit discussion on the generalizability of the method to various image sets.
    • The previous works referenced (e.g., SDEdit) should be briefly described for reader convenience, without necessitating an external search for the referenced paper.

    • There's a deficiency in the literature review concerning image generation based on diffusion probabilistic models, requiring more depth and exploration in this area.
-------------------------------------------------------------------------------------------------------------------------------------
POST-REBUTTAL ASSESSMENT:
The concerns have been addressed in the revised paper.

**Questions:**

INITIAL REVIEW:

• Correct the typo on page 2 after equation (1) where $g$ should be replaced with $\sigma".

    • The rationale behind introducing the methods in section 2.1 needs clarification, establishing their relevance to the current work. Additionally, references for the methods introduced in this section are missing.

    • Define $\bm{\epsilon}_t$ (as observed in equation (4)) since it hasn't been explicitly defined in the paper.

    • Provide further clarification on whether "Gaussian noise" is i.i.d. (independent and identically distributed) or simply refers to AWGN (Additive White Gaussian Noise).

    • Maintain consistency in the use of spaces after full stops and commas.

-------------------------------------------------------------------------------------------------------------------------------------

POST-REBUTTAL ASSESSMENT:
The questions have been addressed in the revised paper.

---

> ### Author Response · Authors · 2023-11-18
>
> **Q.1**: Practical guidance for selecting these parameters remains unclear.
> > The paper's contribution heavily relies on specific parameter selection (e.g., parameters in eq.(14), boundary values in section 3.5, and the interpolation coefficient  in slerp), yet practical guidance for selecting these parameters remains unclear.
>
> ***Ans for Q.1***:
>
> Thank you for pointing out the problem.
>
> - We agree with your point that the performance of our method is heavily related to the introduced hyperparameters. In this context, we would like to note that these hyperparameters are selected from a relatively stable range in our experiments. Specifically,$\gamma$ is within the range $[0, 0.1]$, $\mu$ and $\nu$ are within the range $[0.8, 1]$, the boundary is within the range $[2.0, 2.4]$. Users only need to set $\alpha$ and $\beta=\sqrt{1-\alpha^2-\gamma^2}$ can be automatically calculated.
>
> - We appologize for the missing strategy to select hyperparameters. In our experiments, we select hyperparameters from a subset of images (less than 20 images) and apply these hyperparameters to the remaining images.
>
>
>
> **Q.2**: Computational overhead and potential limitations.
> > Computational overhead and potential limitations of implementing the proposed method in practical scenarios lack detailed explanation.
>
> ***Ans for Q.2***:
>
> Thanks for your constructive suggestions. Accordingly, we have added the following explanation and discussion to our revision.
>
> - Our method involves an extra step compared to using SDEdit for interpolation, which is the mapping of the image to the latent variable. This extra overhead will double our processing time compared to interpolate images using SDEdit. As a result, this extra overhead leads to better feature preservation, as the noise and image information are quite balanced in this case.
> - Regarding potential application limitations, our paper mainly focus on image data. Thus, its effectiveness in other modalities has not been validated, which a potential limitation of our work. We will explore the posibility of our method on different modalities in our future work.
>
> Thanks again for your constructive suggestions, and we believe the above discussion could significantly improve the quality of our paper.
>
> **Q.3**: Further exploration to assess with diverse settings.
> >The experiments are conducted under a simplified scenario where the drift coefficient $\mu(x_t,t)$ is set to zero. Further exploration is needed to assess the level of improvement and applicability in diverse domains or image types, as seen in Figure 6. The paper lacks explicit discussion on the generalizability of the method to various image sets.
>
> ***Ans for Q.3***:
>
>  Thank you very much for your valuable suggestions. Following your advice, we have conducted supplementary experiments and provide explanation to our revision.
>
> - We set $\mu(x_t,t)=0$ following the defaul setting of previous work [1], while overlooking the potential limitation of this experimental setting. Inspired by your valuable comments, we have conducted more experiments using Stable Diffusion, and reported the results in Appendix C.2. Specifically, Stable diffusion is built upon DDPM, and the drift coefficient of DDPM is non-zero. The results demonstrate that our approach performs quite well under various settings.
> - Follwing your valuble suggestion, we further verify the effectiveness of the proposed method on more datasets, including tiger-lion, peach and so on. The results are listed in Apendix C.2 demonstrating that our method can consistently achieve excellent results.
>
> Thanks again for your constrctive suggestions. We have added the above discussion to our revised paper.
>
> [1]Karras, T., Aittala, M., Aila, T., and Laine, S. Elucidating the design space of diffusion-based generative models. In Proc. NeurIPS, 2022.

---

> ### Author Response · Authors · 2023-11-18
> **Supplemental**
>
> **Q.4**: Writing problem
> >①The previous works referenced (e.g., SDEdit) should be briefly described for reader convenience, without necessitating an external search for the referenced paper.
> >②There's a deficiency in the literature review concerning image generation based on diffusion probabilistic models, requiring more depth and exploration in this area.
> ③Correct the typo on page 2 after equation (1) where  should be replaced with $\sigma$".
> ④The rationale behind introducing the methods in section 2.1 needs clarification, establishing their relevance to the current work. Additionally, references for the methods introduced in this section are missing.
> ⑤Provide further clarification on whether "Gaussian noise" is i.i.d. (independent and identically distributed) or simply refers to AWGN (Additive White Gaussian Noise).
> ⑥Define ${\epsilon}_t$ (as observed in equation (4)) since it hasn't been explicitly defined in the paper.
>
> ***Ans for Q.4***:
>
> Thanks for your kind suggestions. Accodingly, we have revised the paper and added explanations as follows.
>
> - Following your valuable suggestion, we have added the brief introduction of the mentioned SDEdit for reader convenience.
>
>   **supplementary content**: The SDEdit accomplishes image modifications by overlaying the desired alterations onto the image, introducing noise, and subsequently denoising the composite. This process ensures that the resulting image maintains a high level of quality.
>
> - We have discussed more works about diffusion probabilistic models in our revision.
>
>   **supplementary content**: Diffusion models generate data by progressively perturbing data into noise through Gaussian perturbations and then creating samples from the noise using sequential denoising steps. To date, diffusion models have also been applied to various tasks, such as image generating(Rombach et al., 2022; Song & Ermon, 2020; Nichol et al., 2021; Jiang et al., 2022), image super-resolution(Saharia et al., 2022c; Batzolis et al., 2021; Daniels et al., 2021), image inpainting(Esser et al., 2021), image editing(Meng et al., 2021), and image-to-image translation(Saharia et al., 2022a). Specifically, latent diffusion models(Rombach et al., 2022) stand out in generating text-conditioned images, garnering widespread acclaim for their ability to produce highly realistic visual images.
>
>
> - We have fixed the typos.
> - We have highlighted the rationale behind the proposed method as follows.
>
>   **supplementary content**: Here we first introduce how to describe the diffusion model's noise injection and denoising process in the form of Stochastic Differential Equations (SDEs). Building upon this, we provide a brief overview of how diffusion models are used for image interpolation and image editing. Through image editing, we can implement an interpolation method that doesn't require latent variables. These methods form the foundation for the approach we propose."
> - We have unified the expression of the utilized noise with independent and identically distributed Gassian noise following Denoising Diffusion Probabilistic Models(DDPM).
>
>   **supplementary content**: The  noise added to the two images can be either the same or different i.i.d Gaussian noise, and we will demonstrate shortly that they exhibit only minor distinctions.
>
> - We have supplemented the definition of $\epsilon_t$.
>
>   **supplementary content**: In this context, we denote the original image as $x_0^{(i)}$, and $x_t^{(i)}$ represents the noised image, corresponding to the variable of the images in the latent space with noise level $\epsilon_t$. Utilizing the probability flow ODE for its stability and unique encoding capabilities, we encode $x_0$ into the latent space by integrating Eq. 3, and we denote this encoding process as a function $f$.

---

> ### Author Response · Authors · 2023-11-20
> **Welcome for more discussions**
>
> Thanks for your valuable time in reviewing and insightful comments. Following your comments, we have tried our best to provide responses and revise our paper. Here is a **summary of our response** for your convenience:
> - (1) **Practical guidance for selecting these parameters**: In this context, we would like to note that these hyperparameters are selected from a relatively stable range in our experiments. Specifically,$\gamma$ is within the range $[0, 0.1]$, $\mu$ and $\nu$ are within the range $[0.8, 1]$, the boundary is within the range $[2.0, 2.4]$. Users only need to set $\alpha$ and $\beta=\sqrt{1-\alpha^2-\gamma^2}$ can be automatically calculated.
> - (2) **Computational overhead and potential limitations**: ①Our method involves an extra step compared to using SDEdit for interpolation, which is the mapping of the image to the latent variable. This extra overhead will double our processing time compared to interpolate images using SDEdit and this extra overhead leads to better feature preservation. ②Regarding potential application limitations, our paper mainly focused on image data. Thus, its effectiveness in other modalities has not been validated.
> - (3) **Further exploration to assess with diverse settings**: We have conducted supplementary experiments and provide explanation to our revision.
> - (4) **Writing problem**: ①We have added the brief introduction of the mentioned SDEdit for reader convenience. ②We have discussed more works about diffusion probabilistic models in our revision. ③We have fixed the typos. ④We have unified the expression of the utilized noise with independent and identically distributed Gassian noise following Denoising Diffusion Probabilistic Models(DDPM). ⑤We have supplemented the definition of $\epsilon_t$.
>
>
> We humbly hope our repsonse has addressed your concerns. If you have any additional concerns or comments that we may have missed in our responses, we would be most grateful for any further feedback from you to help us further enhance our work.

---

> ### Author Response · Authors · 2023-11-21
>
> Thanks a lot for your time in reviewing and reading our response and the revision. We sincerely understand you’re busy. But as the window for responsing and paper revision is closing, would you mind checking our response (a brief summary, and details) and confirm whether you have any further questions? We are looking forward to your reply.

---

> ### Comment · Reviewer_YU98 · 2023-11-21
> **Post-rebuttal Decision**
>
> Thank you for your thorough responses. I've reviewed your replies to my questions and the revised paper. It's evident that you've diligently addressed the issues/ambiguities, and the changes made in the revised paper have contributed to a better understanding of your work and strengthened the paper. Since I'm satisfied with the changes made, I am inclined to reassess the paper positively and recommend acceptance.

---

### Official Review · Reviewer_h8PS · 2023-11-06

**Soundness:** 3 good
**Presentation:** 2 fair
**Contribution:** 3 good
**Rating:** 8
**Confidence:** 3

**Summary:**

This paper addresses image interpolation based on diffusion models; it shows the limitation of the linear spherical interpolation (Song 2020), and then proposes a novel method. Specifically, the proposed method integrates the linear spherical interpolation and the linear spherical interpolation combined with SDEdit; it denoises the linear combination of the latent variables, the original images, and Gaussian noise. The performance of the proposed method is compared with those of the linear spherical interpolation and the linear spherical interpolation combined with SDEdit on bedroom and cat images.

**Strengths:**

First of all, the experimental results qualitatively demonstrate that the proposed method works better than both the linear spherical interpolation and the linear spherical interpolation combined with SDEdit. Second, the proposed method generalizes the linear spherical interpolation and the linear spherical interpolation combined with SDEdit so that it includes both the methods in special cases. Third, the theoretical backgrounds are interesting; the limitation of the linear spherical interpolation is explained by Theorem 1, and the proposed method is supported by Theorem 2.

**Weaknesses:**

First, the proposed method has a number of empirical parameters (alpha, beta, gamma, mu, nu, and boundary). It would be difficult to appropriately set those parameters in practice, although the effects of each parameter is experimentally demonstrated. Second, the experimental evaluation is limited: only qualitative evaluation on small number of images. Third, the presentation of this paper could be improved.
In my opinion, Introduction is too short and the number of references are too small.

**Questions:**

I would be happy to receive your feedback to the comments on Weaknesses.

---

> ### Author Response · Authors · 2023-11-18
>
> We sincerely thank you for taking the time to review.  According to your comments, we provide detailed feedback below and also add them into the revision. We believe that these changes could significantly improve the overall quality of our work:
>
> **Q.1**: It would be difficult to  set the parameters.
> > First, the proposed method has a number of empirical parameters (alpha, beta, gamma, mu, nu, and boundary). It would be difficult to appropriately set those parameters in practice, although the effects of each parameter is experimentally demonstrated.
>
> ***Ans for Q.1***:
>
> Thanks for pointing out this potentially confusing problem. Accordingly, we have supplemented the relevant explanations in the revision.
> The introduced hyperparameters are listed as follows. Although our method introduces several hyperparameters, we would like to note that it is relatively effortless to tune the parameters.
> - i) Boundary determines the extent of noise constraint. In our experiments, the boundary is within the range $[2.0, 2.4]$.
> - ii) $\gamma$ determines the strength of the added noise. In our experiments, $\gamma$ is within the range $[0, 0.1]$.
> - iii) $\mu$ and $\nu$ supplement the information lost in the generation process (by introduing the information from the original image). In our experiments, $\mu$ and $\nu$ are within the range $[0.8, 1]$.
>
> In addition, our method involves a user-determined parameter $\alpha$.
> - $\alpha$ is set by users, since it controls the degree of interpolation bwteen two images, where larger $\alpha$ means the generated image is more similar to the first image. In this context, $\beta = \sqrt{1-\alpha^2-\gamma^2}$.
>
>
> **Q.2**: The experimental evaluation is limited.
> > Second, the experimental evaluation is limited: only qualitative evaluation on small number of images.
>
> ***Ans for Q2***:
>
> Thank you for your constructive comments. Accordingly, we have added the following experiments and explanations to our revisoin.
>
> - We have conducted experiments using more models and datasets. Specifically, we evaluate our method using Stable Diffusion model on  eagle, tree, car and other datasets. The results are reported in the Appendix C.2. These reults show that our method can consistently achieve good performance under various settings.
>
> - We agree that an appropreate quantitative metric is crucial for comparing difference methods. However, designing such a metric is challenging in the literature, since it should not only be based on pixels but also consider the semantic aspect to assess the quality of the interpolated images. The challenge may be the reason why previous methods in the literature did not involve such a metric. In this context, we would like to enhance the persuasiveness of our results by reporting additional results in the appendix. We will leave it as our future work to explore methods for quantifying the quality of interpolated images.
>
> We believe these results and discussion would make our work more solid. Thanks again for your valuable comments.
>
> **Q.3**: The introduction and the references is too short.
> > Third, the presentation of this paper could be improved. In my opinion, Introduction is too short and the number of references are too small.
>
> ***Ans for Q.3***:
>
> Thanks for your kind suggestion. Accodingly, we have modified the paper to provide a detailed introduction and added more references. The corresponding content will be placed in the next response box.

---

> ### Author Response · Authors · 2023-11-18
> **Supplemental information for the introduction content.**
>
> **supplementary content**:
>
> Image interpolation is an exceptionally fascinating task, not only for generating analogous images but also for igniting creative applications, especially in domains like advertising. At present, state-of-the-art generative models showcase the ability to produce intricate and captivating visuals, with many recent breakthroughs deriving from diffusion models. (Ho et al., 2020; Song et al., 2020a; Rombach et al., 2022; Saharia et al., 2022b; Ramesh et al., 2022). The potent of diffusion models is widely acknowledged, but to our knowledge, there has been relatively little research on image interpolation with diffusion models.(Croitoru et al., 2023)
>
> Within the realm of diffusion models, the prevailing technique for image interpolation is linear spherical interpolation(Song et al., 2020a;b). This approach shines when employed with images generated by diffusion models. Nonetheless, when extrapolated to images not originating from diffusion models, the quality of interpolation outcomes might fall short of expectations and frequently introduce substantial artifacts.
>
> We initially analysed the image interpolation process and attributed subpar interpolation results to the introduction of noise with image-related information. This particular noise was not aligned with the level of denoising, resulting in artifacts in the final interpolated images. Directly manipulating the mean and variance of noise through translation and scaling is a very straightforward approach to bring them closer to the desired distribution. However, this not only fails to improve the quality of the image but also results in the loss of a significant amount of original image information. Based on this analysis, we integrated the SDEdit(Meng et al., 2021) method, proposing the substitution of this noise component with random Gaussian noise. While this approach improves image quality, it comes at the expense of introducing additional information.
>
>
> After that, we further devised an innovative interpolation technique that leverages the strengths of both methods. This approach retains valuable noise while introducing subtle Gaussian noise to enhance the quality of interpolation. Furthermore, we’ve introduced a new constraint on the noise component responsible for generating artifacts and incorporated original image to supplement missing information. These improvements not only enhance interpolation results for images within the training domain but also extend the capability to interpolate with natural images outside the training domain, yielding the best interpolation results achieved to date. In light of the limited exploration in this domain in previous studies(Croitoru et al., 2023), we aspire that our research can serve as a source of inspiration for fellow researchers.

---

> ### Author Response · Authors · 2023-11-20
> **Welcome for more discussions**
>
> Thanks for your valuable time in reviewing and insightful comments. Following your comments, we have tried our best to provide responses and revise our paper. Here is a **summary of our response** for your convenience:
> - (1) **The setting of the parameter**: It is relatively effortless to tune the parameters. For users, it is only necessary to set the parameter $\alpha$ and ensure that other parameters are within the selected range to generate high-quality images
> - (2) **Experimental evaluation**: We have conducted experiments using more models and datasets. And designing a metric is challenging, since it should not only be based on pixels but also consider the semantic aspect to assess the quality of the interpolated images.
> - (3) **The introduction and the reference**: We have modified the paper to provide a detailed introduction and added more references.
>
> We humbly hope our repsonse has addressed your concerns. If you have any additional concerns or comments that we may have missed in our responses, we would be most grateful for any further feedback from you to help us further enhance our work.

---

> > ### Comment · Reviewer_h8PS · 2023-11-23
> >
> > Thank you very much for your response and revision. Because the response is convincing to me, I would like to change my rating from 6 (marginally above the acceptance threshold) to 8 (accept, good paper).

---

### Meta-Review · Area_Chair_xkYT · 2023-12-13

**Metareview:**

Traditional image interpolation doesn't work for real images which are not generated by diffusion models. The paper analyzed the reason and attributed it to the introduction of noise with image-related information. This noise is not aligned with the level of denoising, leading to artifacts in the interpolated images. The proposed method integrates the linear spherical interpolation and the linear spherical interpolation combined with SDEdit. It denoises the linear combination of the two latent variables, the original images and the Gaussian noise, which outperfoms previous methods.
Strengths: (1) theoretical analysis is provided, (2) the paper identifies the limitations of previous methods in handling natural images, and (3) proposes a solution.
Weaknesses: (1) the method has multiple empirical parameters which are hard to tune, (2) limited experiment evaluation, (3) the paper can be improved.
The camera-ready paper should address the weaknesses.

**Justification For Why Not Higher Score:**

The paper should address the weaknesses raised by the reviewers, especially paper writing, how to tune the parameters, and more experiment evaluations.

**Justification For Why Not Lower Score:**

The paper should be promoted given that it analyzed and addressed a very interesting problem.

---

### Decision · Program_Chairs · 2024-01-16

Accept (spotlight)